



# On the links between sub-seasonal clustering of extreme precipitation and high discharge in Switzerland and Europe

Alexandre Tuel[1,2], Bettina Schaefli[1,2], Jakob Zscheischler[2,3,4], and Olivia Martius[1,2,5]

[1]Institute of Geography, University of Bern, Switzerland
[2]Oeschger Centre for Climate Change Research, University of Bern, Switzerland
[3]Department of Computational Hydrosystems, Helmholtz Centre for Environmental Research - UFZ, Leipzig, Germany
[4]Climate and Environmental Physics, University of Bern, Switzerland
[5]Mobiliar Lab for Natural Risks, University of Bern, Switzerland

**Correspondence:** Alexandre Tuel (alexandre.tuel@giub.unibe.ch)

**Abstract.** River discharge is impacted by the sub-seasonal (weekly to monthly) temporal structure of precipitation. One example is the successive occurrence of extreme precipitation events over sub-seasonal timescales, referred to as temporal clustering. Its potential effects on discharge have received little attention. Here, we address this question by analysing discharge observations following extreme precipitation events either clustered in time or occurring in isolation. We rely on two sets of precipitation and discharge data, one centered on Switzerland and the other over Europe. We identify "clustered" extreme precipitation events based on the previous occurrence of another extreme precipitation within a given time window. We find that clustered events are generally followed by a more prolonged discharge response with a larger amplitude. The probability of exceeding the 95th discharge percentile in the five days following an extreme precipitation event is in particular up to twice as high for situations where another extreme precipitation event occurred in the preceding week compared to isolated extreme precipitation events. The influence of temporal clustering decreases as the clustering window increases; beyond 6-8 weeks the difference with non-clustered events is negligible. Catchment area, streamflow regime and precipitation magnitude also modulate the response. The impact of clustering is generally smaller in snow-dominated and large catchments. Additionally, particularly persistent periods of high discharge tend to occur in conjunction with temporal clusters of precipitation extremes.

## 1 Introduction

Extreme precipitation accumulations over relatively short sub-seasonal time windows can increase water levels in rivers and lakes, and consequently lead to floods. Such accumulations can result from persistence in precipitation, either as moderate precipitation stretching over many successive days, or as several extreme precipitation episodes separated by a few days or weeks (Merz and Blöschl, 2003), referred to as temporal clusters of extreme precipitation (TCEP) (Kopp et al., 2021; Tuel and Martius, 2021a). The accumulated precipitation brings soils to saturation, preventing subsequent precipitation from infiltrating into the soil and directing it instead to river or overland flow (Paschalis et al., 2014). Considering that extreme precipitation events can lead to flash floods (Doswell et al., 1996), mass movement (Guzzetti et al., 2007; Panziera et al., 2016) or landslides (Bevacqua et al., 2021a), their temporal clustering and the associated soil moisture increase may exacerbate these impacts.





TCEP, which can be considered temporally compounding events (Zscheischler et al., 2020), were linked to several major floods across Europe over the last few decades: in Central Europe during summer 2013 (Grams et al., 2014), in the UK during

winter 2013/2014 (Priestley et al., 2017), or in Switzerland in the fall of 1993, 2000 and 2002 (Barton et al., 2016). The recent Western European floods of summer 2021 were also associated with the successive occurrence of several extreme precipitation events, from mid-June to mid-July (Kreienkamp et al., 2021).

Quantifying the relevance of TCEP for high discharge levels is therefore important to properly characterise flood risk, improve forecasts, support process-based calibration of rainfall-runoff models (Cullmann et al., 2008; Brunner et al., 2021) and develop

informed storylines for impact assessment (Sillmann et al., 2021). The impact of sub-seasonal TCEP on discharge has not been explicitly addressed, to our knowledge, except briefly in the case of Switzerland by Tuel and Martius (2021b). They argued that TCEP increased the likelihood and duration of high discharge events compared to precipitation extremes occurring in isolation.

The study of the influence of the temporal structure of precipitation on the catchment-scale hydrologic response is one of the foundations of flood response and flood frequency analysis. Accordingly, there is an important body of literature studying the

theoretical interplay of temporal and spatial rainfall structure on the streamflow response (e.g., Rodriguez-Iturbe and Eagleson, 1987; Woods and Sivapalan, 1999; Viglione et al., 2010; Zhu et al., 2018). Past work has however generally focused on single events. Similarly, empirical or numerical analyses of observed events often analyze how discharge is affected by precipitation over short time windows only (e.g., Paschalis et al., 2014; Froidevaux et al., 2015; Keller et al., 2018). Additionally, antecedent soil moisture is one of the key controls on the streamflow response to rainfall and on flood generation (Blöschl et al., 2017;

Berghuijs et al., 2019). Antecedent soil moisture strongly modulates the influence of the temporal structure of intense precipitation on the discharge response (Nied et al., 2014, 2017; Keller et al., 2018), and it is itself influenced by the temporal structure of precipitation. High antecedent soil moisture, which favours a larger discharge response, typically results from long-duration precipitation (which can involve TCEP events), but sometimes also from snowmelt (Berghuijs et al., 2019).

In parallel, several studies investigated the tendency for extreme precipitation to cluster in time over sub-seasonal timescales,

from regional scales (Barton et al., 2016; Yang and Villarini, 2019; Tuel and Martius, 2021b) to global scales (Kopp et al., 2021; Tuel and Martius, 2021a), as well as the role of TCEP in extreme precipitation accumulations. Over Europe, possible drivers of TCEP include temporal clustering in extratropical cyclones, particulary in winter (Mailier et al., 2006; Vitolo et al., 2009; Pinto et al., 2013), persistence in large-scale teleconnection patterns (Yang and Villarini, 2019), recurrent Rossby Waves (Ali et al., 2021), tropical forcing and blocking (Barton et al., 2016). The explicit impacts of TCEP on discharge and floods

have however mainly been discussed for case studies of major flood events only. Barton et al. (2016) discovered that TCEP caused the Lake Maggiore floods of fall 1993 and 2000, by bringing large amounts of precipitation at intervals that were too short for the lake level to decrease between events. The Central Europe floods of summer 2013 (Grams et al., 2014) as well as the UK floods of winter 2013/2014 (Priestley et al., 2017) were also connected to TCEP. Tuel and Martius (2021b) conducted a more systematic analysis on the relationships between TCEP and high discharge over Switzerland. They found in particular

that TCEP led to a higher probability of high discharge than non-clustered precipitation extremes.

Here, we build on the methodology of Tuel and Martius (2021b) to quantify the effects of TCEP on discharge in Switzerland and Europe, specifically on the occurrence and temporal persistence of high discharge. Tuel and Martius (2021b) only looked at





3-week TCEP across Swiss catchments. Here, by contrast, we classify extreme precipitation events according to their clustering timescale, and analyse the sensitivity of results to that timescale, as well as to catchment area and to extreme precipitation magnitude. We also extend the scope to Europe, to explore a wider range of catchment characteristics (climate, area, etc.). Our analysis relies on two sets of precipitation and discharge data, one for Switzerland, also used by Tuel and Martius (2021b), and one for Europe. We take a forward and backward approach (e.g. Zscheischler et al., 2014), analyzing on the one hand the characteristics of discharge following clustered and non-clustered extreme precipitation events, and on the other hand the frequency of TCEP preceding particularly persistent high discharge periods.

## 2 Data and methods

### 2.1 Data

#### 2.1.1 Catchments and discharge data

We use two discharge datasets in this study. The first consists of daily discharge data for 96 small to medium-sized gauged catchments (14-1700 km$^2$, with an average area of 294 km$^2$) distributed across Switzerland (Figure 1-a). These catchment cover most of Switzerland's climates and range in mean elevation from 500 to 2700 masl. The data is collected and distributed by Switzerland's Federal Office for the Environment (FOEN). We selected catchments among all available ones based on several criteria: at least 10 years of common data coverage with the precipitation data (RhiresD, 1961-2019, see section 2.1.2), no major lakes, no significant human influence on discharge, and no detected non-stationarity in annual discharge maxima series (as determined by the FOEN). This set of catchments, with a few exceptions, was also used by Muelchi et al. (2021) and Tuel and Martius (2021b).

Daily discharge data for Europe comes from the Global Runoff Data Center dataset (GRDC). We selected all available gauges (amounting to 637 in total) in the 12°W-38°E/28°N-72°N domain. We required catchments to have an area of 50,000 km$^2$ or less to exclude very large catchments and a minimum of 10 years of overlap between discharge and precipitation data (EOBS, 1950-2019, see section 2.1.2). In the end we retained 500 catchments, ranging from 9 to 2886 masl in mean elevation (calculated from 30 arc-second GTOPO30 data) and 10 to 48,550 km$^2$ in area (Fig. 1-b).

To focus on the fast (daily to weekly-scale) discharge response to precipitation extremes, we remove the baseflow component with the recursive digital filter algorithm of Nathan and McMahon (1990) (with a filter parameter of 0.925) implemented in the R EcoHydRology package.

#### 2.1.2 Precipitation

Reference precipitation data for Switzerland come from the daily 2x2km RhiresD dataset, available from 1961 to present. We use data until 2019 only. RhiresD is obtained by spatially interpolating data from a high-density station network covering the whole of Switzerland. Additional details on this dataset can be found on the dedicated MeteoSwiss web page at https://www.meteoswiss.admin.ch/home/climate/swiss-climate-in-detail/raeumliche-klimaanalysen.html. For European precipitation, we





use the daily EOBS version 21.0e dataset at 0.25° resolution (Haylock et al., 2008) over the 1950-2019 period. EOBS is also the result of a spatial interpolation of measurements across the station network of the European Climate Assessment & Dataset (ECA&D). The gridded precipitation data are averaged for each catchment: RhiresD for all FOEN catchments across Switzerland, and EOBS for all GRDC catchments across Europe, yielding two precipitation datasets.

## 2.2 Methods

For each catchment, we conduct the analysis over the period for which both discharge and precipitation data are available. This means that daily discharge and precipitation percentiles are calculated over different time periods depending on the catchment.

### 2.2.1 Precipitation and discharge extremes

As in Tuel and Martius (2021b), for each catchment, we define precipitation extremes on a monthly basis as days when catchment-averaged precipitation exceeds its $99^{th}$ percentile for the corresponding month. All January days are thus compared to the January $99^{th}$ percentile of daily accumulated precipitation. This removes the seasonal dependence in extreme precipitation magnitude and leads to a constant rate of extreme precipitation occurrence across the year. This step is motivated by the fact that high discharge is shaped not only by precipitation, but also by surface conditions like snow and vegetation cover, soil saturation or evaporative demand (Paschalis et al., 2014; Nied et al., 2017). Consequently, the seasonal cycles of extreme precipitation and discharge magnitude often differ significantly, with the highest discharge not necessarily occurring after the heaviest precipitation events (Tuel and Martius, 2021b).

The persistence of individual weather systems over timescales of 1-2 days leads to short-term dependence in the occurrence of extreme precipitation events. To remove this dependence, we apply a runs declustering procedure (Coles, 2001) in which extreme events separated by less than two days (Barton et al., 2016; Tuel and Martius, 2021b) are merged into a single event. For each catchment, precipitation extremes are then classified into different categories based on their degree of temporal clustering (Figure 2-a). We focus on different sub-seasonal timescales of temporal clustering: events which occurred between $n-1$ and $n$ weeks after another event are put into the "$n$-week" category, where $n \in \{1, 2, 3, 4, 5, 6, 7, 8\}$. All remaining events are declared as "non-clustered" and put together in a separate category. For simplicity, and also because it only has a minor effect on the results, we analyse the $n = 5$ and $n = 6$ categories together, as well as the $n = 7$ and $n = 8$ categories. Results will thus be shown for $n \in \{1, 2, 3, 4, 6, 8\}$. Note that these categories do not intersect: each extreme event belongs to one and one only.

We use the $95^{th}$ percentile of daily discharge to define high discharge days for all catchments. Hereafter we will not mention the $95^{th}$ percentile and simply refer to "high discharge" for simplicity. Unlike for precipitation, this percentile is fixed throughout the year and calculated on the entire available time series, because impacts of discharge extremes are more related to their absolute rather than their relative magnitude. We choose a smaller percentile threshold compared to precipitation to capture the majority of high discharge events associated with extreme precipitation events because discharge is influenced by factors other than precipitation. Potential long-term trends in extreme daily precipitation or discharge percentiles are not taken into account.



### 2.2.2 Effects of temporal clustering in extreme precipitation on discharge

We quantify the effect of temporal clustering of precipitation extremes on discharge by considering several simple metrics. For each catchment and each clustering category, we calculate:

1. daily discharge percentiles;

2. daily high discharge probabilities;

3. daily high discharge odds ratios

for each of the 30 (Switzerland data) or 60 (GRDC data) days, averaged across all extreme precipitation events in each clustering category. We limit the analysis to 30 and 60 day windows respectively because the response to precipitation extremes is largely confined to these windows. The odds ratio compares the odds of high discharge given the occurrence of a precipitation extreme to the odds of high discharge given the absence of a precipitation extreme. With $p_1$ the probability of high discharge given that a precipitation extreme occurred, and $p_2$ the probability of high discharge given that a precipitation extreme did not occur, the odds ratio is equal to by $\frac{p_1(1-p_2)}{p_2(1-p_1)}$ (Wilks, 2019). It measures the strength of the link between the occurrence of extreme precipitation and that of high discharge, but not its absolute magnitude. For the latter, it is more relevant to consider the high discharge probability after precipitation extremes.

We also calculate mean high discharge probability and odds ratio values over a 5-day window following extreme precipitation events. This time window captures the bulk of the discharge response to extreme precipitation for almost all catchments (see Results section), though our results remain approximately the same for timescales between 3 and 10 days. Finally, we define an high discharge response timescale as the time window during which the probability of high discharge continuously remains above 10% in the 30/60 days following an extreme precipitation event (Figure 2-b). This probability is exceeded on at least one day after extreme precipitation events for almost all catchments, and so the response timescale is almost always longer than 1 day. The 10% threshold may seem small, but it still corresponds to a doubling of the baseline probability. In addition, because the data have a daily resolution, we can only detect differences in high discharge response timescales, between clustered and non-clustered precipitation extremes, of at least one day. Hence we need to select a threshold low enough so that the difference will be at least one day (otherwise it would not be detected). For these reasons we selected the 10% threshold.

### 2.2.3 Persistent high discharge periods and precipitation characteristics

We identify periods of persistent high discharge following Tuel and Martius (2021b). Daily discharge series (with the baseflow removed) are first translated to binary series in which non-zero values correspond to high discharge days (when discharge exceeds its 95$^{th}$ percentile). We then look for $L-day$ periods with at least $N$ non-zero days where $(L, N) \in \{(10, 1), (10, 5), (20, 10), (40, 20)\}$. We proceed as follows (Figure 2-c): starting with the largest $L$ value ($L = 40$), we apply an $L - day$ moving average to the binary series and select the period with the largest event total. The beginning of the period is defined as the first high discharge day within the $L$-day window. Non-zero values during that period are then set to zero, and the search is repeated as long as new periods are found. We then move on to the next largest $L$ value, and repeat the process. The procedure ensures that all





identified periods belong to only one $(L, N)$ category. Note that depending on the values of $L$ and $N$, no periods may be found in some catchments.

To characterise precipitation before and during the persistent high discharge periods, we then calculate for each catchment
and each $(L, N)$ the average cumulative precipitation percentile and number of extreme precipitation events over various time windows: 0-2 days, 3-7 and 7-21 days before the events, as well as during the events themselves: from day 0 (beginning of events) to day $L - 1$. This choice of time windows follows Froidevaux et al. (2015) who analysed the distribution of precipitation before annual discharge peaks across Switzerland. The cumulative precipitation percentiles are calculated with respect to all periods of the same length within $\pm 20$ calendar days of observed persistent high discharge periods.

## 3   Results

### 3.1   Effects of temporal clustering in extreme precipitation on discharge

#### 3.1.1   Switzerland

We begin with the results for the selected Swiss catchments. Table 1 indicates the average number of extreme precipitation events in each clustering category. Non-clustered events (*i.e.*, not preceded by another event in the previous 8 weeks) account
for about 60% of all extreme precipitation events, while each subsequent category represents between 5 and 10% of events. The sample is thus much larger for non-clustered events, and we expect less variability in the corresponding results.

We show the per-day distribution of discharge percentile, high discharge probability and odds ratio after clustered and non-clustered extreme precipitation events, averaged across catchments with a mean elevation lower than 1500m (Fig. 3) and higher than 1500m (Fig. 4). We separate Swiss catchments into these two groups because discharge in high-elevation catchments is
typically snow- or glacier-dominated, and we expect their discharge to exhibit thereby a different sensitivity to precipitation extremes.

The discharge response is mainly confined to the first 20 days following the extreme precipitation event (day 0) (Fig. 3). Peak response occurs on day 1, with average discharge percentiles of 0.9 and higher. Most catchments already exhibit a substantial response on day 0, likely because of their relatively small size (on average 300 km$^2$). After the peak, discharge slowly recedes
back to its baseline level (50$^{\text{th}}$ percentile) reached on average after 30 days (Fig. 3-a).

Clustering generally enhances the discharge response. It leads to a higher discharge peak (Fig. 3-a) and high discharge likelihood (Fig. 3-b) on day 1, and to a larger discharge response afterwards. Clustering at the 1-2 week window has the strongest impact on discharge, and the influence of clustering weakens as the window increases (Fig. 5). During the first 5 days in particular, the probability and odds ratio of high discharge are significantly larger for 1-week clustered events than non-clustered
events (Figs. 3-d,e and 5) (where the significance of the difference between responses to clustered and non-clustered extremes is assessed with a two-sample t-test). The peak odds ratio is notably more than twice as large on average, and the 1-5 day odds ratio is almost twice as large (Fig. 5-b). Even for the 4-week clustering timescale, the odds ratio remains 30% larger than in the non-clustered case (Fig. 3-c). Discharge after clustered events remains higher than after non-clustered ones for at least 10 days,





which translates into longer response timescales (as defined in section 2.2.2). The probability of high discharge remains above

0.1 for an average of four days after non-clustered events but for more than five days for clustered events in the cluster length categories up to four weeks (Fig. 5-c). Results for the eight-week clustering window are generally indistinguishable from the non-clustered category.

Regardless of the clustering category, the discharge response is weaker at high elevations (Fig. 4). Extreme daily discharge values are much less common right after precipitation events: peak probabilities and odds ratio are reduced by a factor 2-3

compared to the low-elevation catchments. The impact of clustering is also less pronounced when compared to the non-clustered category (Fig. 4-d,e,f). The peak response still occurs on day 1, but day 0 values are proportionately higher than at low elevations (compare first rows of Figs. 3 and 4). High-elevation catchments are, on average, not much smaller than low-elevation catchments, but they have a less dense vegetation cover, with shallower soils and steeper slopes, which might lead to a quicker onset of overland flow (via infiltration excess or saturation excess) as well as to faster subsurface flow (Carrillo et al.,

2011) and thus explain the faster discharge response.

Because clustered event categories contain on average substantially fewer events than the non-clustered one (Table 1), associated results generally exhibit more variability. Additionally, given that Switzerland covers a relatively small area, the same heavy precipitation events often affect several catchments at the same time. The samples used to obtain the curves in Figs. 3 and 4 are thus not independent. We notice for instance an increase in average discharge percentiles (and also in high discharge

probability) around days 15-25 for the 1- and 4-week categories on Fig. 3-a,b. They result from heavy precipitation that occurred simultaneously over many catchments after a few events in the 1- and 4-week categories, and are not a delayed response to the initial extreme precipitation on day 0.

    The influence of clustering on discharge extremes also varies in space, beyond the effect of elevation (Figs. 6 and 7). Some regions, like the Jura (northwest) or Southern Switzerland, exhibit larger 1-5 day high discharge probabilities than others with

similar elevation. This holds to some extent already for non-clustered extremes, but is more striking at clustering timescales of 1 and 2 weeks (Fig. 6). The difference in high discharge probability between non-clustered and 1-week clustered events is even statistically significant for several catchments in these two regions, despite the small event number in the 1-week category (Fig. 6-b). For the Jura, this particular regional effect can certainly be related to karst effects (Malard et al., 2016).

    In terms of odds ratio, regional contrasts unrelated to elevation differences are less prominent. The largest odds ratios are found

over much of Northern Switzerland, including the Jura, but no so much in Southern Switzerland (Fig. 7).

### 3.1.2   Europe

Results for the European-wide data are consistent with the ones over Switzerland (Figs. 8 and 9). Because the distribution of catchment areas has a much wider range, and the average catchment area is larger than in the Swiss FOEN dataset, the recession timescales are on average much longer than on Figs. 3 and 4, and the response timescales as well (Fig. 9-c). The

magnitude of the maximum discharge response is smaller, for all considered metrics. In particular, the peak probability of high discharge is about 0.35 after non-clustered events (0.55 after 1-week clustered events) (Fig. 8-b), compared to 0.65 (and 0.9) in the Swiss results (Fig. 3-b). Yet, because the response lasts on average longer, 1-5 day average high discharge probabilities





and odds ratios are similar to the Swiss values (Fig. 5-a,b and Fig. 9-a,b).

Extreme event categories contain, on average, about as many events in the European data as they do in Swiss data. However,
since we average over five times more catchments, and because the precipitation and discharge series of these catchments are
more independent than in the Swiss data, the curves are overall smoother (compare Fig. 3-a,b,c and Fig. 8-a,b,c). The influence
of clustering is also more strictly decreasing with increasing length of the clustering window, whereas results for Switzerland
showed some discrepancies – for instance the discharge response being larger after 6-week than after 4-week clustered events
(Fig. 3-a,b,c).

We do not investigate the influence of elevation in the European data; first, because it covers a much narrower range of eleva-
tions (only 10 catchments have a mean elevation higher than 1000m); second, because mean elevations are less representative
of the elevation distribution in larger catchments; and third, because unlike in Switzerland, the presence of snow is dictated by
other catchment characteristics (chiefly latitude).

Fig. 10 shows the spatial variability of the results. Although GRDC catchments are far from covering all of Europe, we can
see some general tendencies. The discharge response to extreme events, whether clustered or non-clustered, is proportionately
weaker at higher latitudes (Scandinavia) where snowmelt-driven floods are more common (Berghuijs et al., 2019). The largest
high discharge probabilities are found over Central Europe, the British Isles and, to some extent, the Iberian Peninsula.

### 3.2 Persistent high discharge periods and temporal clustering

We now turn to the analysis of precipitation before and during high discharge periods. Most high discharge periods, whether
persistent or not, are preceded by intense precipitation (90th percentile or higher) in the three days before the start of the episode
(Fig. 11-a,b,c). Cumulative precipitation totals are however larger before persistent periods, except at high elevations where
they are less significant than before non-persistent periods. The increase is largest for the most persistent periods ($L = 40$,
$N = 20$), especially in the Jura and Southern Switzerland where values larger than the 98th percentile are found (Fig. 11-c).
Note that although we select a 90% significance level in Fig. 11, precipitation accumulations lower than the percentile can
still be significant, because we assess the significance by comparing to periods at the same time of the year as the periods of
analysis.

Event precipitation is by contrast very different between persistent and non-persistent periods (Fig. 11-d,e,f). Weak and non-
significant precipitation accumulations characterise non-persistent periods, whereas persistent periods, except at high ele-
vations, are associated with intense and significant event precipitation totals, again particularly over the Jura and Southern
Switzerland for the most persistent events (Fig. 11-f).

Intense precipitation accumulations both before and during persistent high discharge periods often result from TCEP (Fig.
11-h,i). More than half of $(L, N) = (40, 20)$ periods in 22 catchments of Northwestern and Southern Switzerland are associ-
ated with TCEP (Fig. 11-i). Typically, one precipitation extreme occurs in the first three days before the event, and another,
sometimes more, occurs during the event itself. Overall, the connection to TCEP is weaker for less persistent high discharge
periods. Admittedly, the time window used to calculate TCEP frequency depends on the value of $L$, and higher TCEP frequen-




cies should be expected for larger values of $L$. Nevertheless, $L$ is the same between non-persistent (Fig. 11-g) and the shorter persistent (Fig. 11-h) events, and TCEP frequencies are overall larger for the latter.

Results for Europe are qualitatively similar (Fig. 12). Intense precipitation accumulations often precede high discharge periods (Fig. 12-a,b,c), and accompany persistent high discharge periods but not the non-persistent periods (Fig. 12-d,e,f). Similarly, TCEP is most often detected during long, persistent periods, and absent from the shorter events (Fig. 12-g,h,i). Overall, fewer significant values are detected compared to Switzerland, but this may possibly result from the larger average catchment size. The probability of high discharge in large catchments is more sensitive to the exact timing and location of extreme precipitation, and catchment-average precipitation series as we use here may be less relevant. We notice in particular that the smallest catchments, located mainly in the British Isles and Central Europe, generally exhibit significant accumulations. Catchments across Scandinavia also exhibit few if any significant values.

## 4 Discussion

Our analysis makes the case for a significant influence of temporal clustering in extreme precipitation on the likelihood and temporal persistence of discharge extremes. Clustered precipitation extremes are followed, on average, by higher discharge values that persist over longer periods than non-clustered precipitation extremes. The shortest clustering timescales (1-2 weeks between successive precipitation extremes) appear to have the most impact, with the influence of clustering progressively decreasing as the timescale increases. Although we subtracted the baseflow component from the discharge series, our results remain very similar if the full discharge series are considered (Fig. A6). The main difference is that response peaks tend to be slightly lower when baseflow is included, and that part of the long-term response to extreme precipitation is also removed when subtracting the baseflow.

Understanding the relationship between precipitation clustering and discharge extremes is important as precipitation clustering characteristics are expected to be affected by climate change (Tuel and Martius, 2021a). For wintertime in a warmer climate in Europe, for instance, we expect an increase in cumulative precipitation from clusters that are however composed by fewer extreme precipitation events in each cluster (Bevacqua et al., 2020).

A detailed process-based analysis is beyond the scope of this paper; nevertheless, we propose that soil moisture memory at sub-seasonal to seasonal timescales (Wu and Dickinson, 2004; Seneviratne et al., 2006) plays an important role in explaining the effects of clustering on discharge. The role of soil moisture pre-conditioning for the likelihood of extreme runoff and discharge has indeed been discussed extensively (Nied et al., 2014; Paschalis et al., 2014; Nied et al., 2017). Soil moisture increases following the first extreme precipitation event, and the short window of time to the next event is not sufficient for soil moisture to decrease back to its initial value. The runoff coefficient during the subsequent extreme event is thus higher, which increases the likelihood of high discharge. A longer period between events means more opportunity for soil moisture to decline, hence the weakening effect of clustering as the clustering window increases (*e.g.*, Fig. 3). TCEP can therefore lead to particularly persistent high discharge periods (Figs. 11, 12).





## 4.1 Snow-dominated catchments

Clustering has a significant impact on discharge for the vast majority of analysed catchments, covering a wide range of spatial
scales and hydroclimates, with the notable exception of high-elevation catchments in Switzerland and high-latitude catchments
in the European data where the effects of precipitation extremes – let alone clustering – on discharge are seldom significant.
In Switzerland, elevation is a direct proxy for the influence of snow and glacier melt. High discharge at high elevations occurs
primarily in summer, at the time of maximum snow- and glacier melt (Figs. 1-a and A2-c). Similarly, high discharge in
Scandinavian catchments occurs most often in conjunction with spring snowmelt (Blöschl et al., 2017; Berghuijs et al., 2019)
(Fig. A4), whereas extreme precipitation magnitude is relatively small in this season compared to summer and fall (Fig. A3).
This does not imply that precipitation extremes and TCEP have no influence on discharge, but rather that their influence is
masked by the seasonality in high discharge, dominated by snowmelt. To detect that influence, "local" discharge percentiles
could be used instead, in the same way as precipitation. The interpretation in terms of impacts would nonetheless be different.
Note also that the GRDC dataset includes almost no catchments along the Norway coast, where floods are less driven by
snowmelt and more by extreme precipitation events (Hegdahl et al., 2020; Berghuijs et al., 2019). The link between discharge
and TCEP in such catchments would thus probably be much higher than for the snow-dominated catchments in the rest of
Scandinavia.

## 4.2 Extreme precipitation magnitude

To define precipitation extremes, we chose monthly-varying percentiles, and set aside the issue of their magnitude. So far, we
indeed analysed extreme precipitation events regardless of their magnitude. One can wonder however how our results change
when taking extreme precipitation magnitude into account. A simple way to tackle this is to separate extreme precipitation
events into two groups based on their absolute magnitude. Pooling clustered and non-clustered extreme events together, the
discharge response clearly scales with the magnitude of the precipitation (Fig. 13-a). This tendency is found in all cluster-
ing groups, but the difference is smaller, in relative terms, for the 1- and 2-week categories than for others (Fig. 13-b). It
cannot simply be explained by differences in precipitation magnitudes between clustering categories (Fig. A5). A detailed
understanding would require taking into account the seasonality in extreme precipitation magnitude and clustering frequency
(Tuel and Martius, 2021b). The largest events generally occur in spring and summer, when surface conditions are often less
conducive to high runoff coefficients (extensive vegetation cover, large evapotranspiration) than in winter (reduced vegetation
cover, frozen/saturated soils). Likewise, for some catchments, TCEP events occur in the season with the largest precipitation
extremes, like in Southern Switzerland, which can bias the result.

## 4.3 Catchment area and response timescale

Another aspect of the results which we did not explore is the influence of catchment area. Given the small range of catchments
areas (14-1700 km$^2$) in the Switzerland dataset, the daily resolution of the discharge data is too coarse to detect a significant
effect. The Europe-wide GRDC dataset, by contrast, covers a much wider range of catchment areas (10-50,000 km$^2$). Again, a





simple approach to the problem is to separate catchments between "small" and "large" ones. We use an arbitrary 10,000 km$^2$ threshold to make the distinction; this leaves 417 catchments classified as "small" and 83 as "large". In large catchments, the peak discharge response to extreme precipitation events occurs later (by 1-2 days on average) than in small catchments, and tends to be lower (Fig. 14-a). Recession timescales are also longer and, as a consequence, high high discharge probabilities persist for much longer (Fig. 14-b). All clustering categories show the same pattern. There is quite a lot of variability in the

results across catchments but, overall, only small catchments have large odds ratios within the first two days following an extreme precipitation event, while a few days later odds ratios in large catchments are bigger (Fig. 14-d). Large catchments have a wider distribution of travel times to the outlet, which smooths the discharge response and leads to a smaller peak on average (Fig. 14-a). In addition, precipitation extremes in large catchments are less likely to extend over the whole catchment than in small ones. They are therefore less likely to drive high discharge in catchments with an area beyond 1000 km$^2$. The soil

moisture memory argument is also less valid for clustered extremes, since two extreme events may occur over different parts of the catchment. However, precipitation extremes are expected to have a larger spatial footprint in a warmer climate (Bevacqua et al., 2021b), such that also larger catchments might experience very fast response times in the future.

## 4.4 Link to high discharge processes

Temporal clustering in extreme precipitation generally leads to a larger discharge response over a longer period of time com-
pared to non-clustered events. Still, the impact of clustering on discharge vary significantly across catchments in both the Switzerland- and the Europe-wide data (Figs. 6 and 10). At first order, the influence of snow and catchment area can explain some of this variability. Yet, in Switzerland at least, some differences do not seem related to catchment elevation or area. Like Tuel and Martius (2021b), we find that the effects of clustering are larger in Northwestern and Southern Switzerland (Fig. 6-b,c). These two regions are already more sensitive to non-clustered extreme precipitation events (Fig. 6-a). Extreme pre-
cipitation magnitude is by far the highest in Southern Switzerland, regardless of the season (Frei and Schär, 1998; Umbricht et al., 2013; Piaget, 2015). It reaches its peak in the fall, which coincides with peak discharge as well (Figs. A1 and A2). High discharge in this region is thus likely driven by infiltration excess (Aschwanden and Weingartner, 1985; Helbling et al., 2006; Diezig and Weingartner, 2007), even in the absence of clustering. The large response to clustering (Fig. 6-b,c) in this region may then simply reflect the larger magnitude of precipitation extremes.

The situation is different in Northwestern Switzerland (Jura mountains). There, high discharge occurs primarily during winter (Fig. A2-a), in conjunction with frozen or saturated soils (Aschwanden and Weingartner, 1985; Helbling et al., 2006) but not with the largest precipitation extremes (Fig. A1-a). Average catchment elevation is rather low (Fig. 1-a), and liquid precipitation and snowmelt not uncommon in winter. The Jura is a region that shows strong karst effects (Malard et al., 2016), which are known to lead to complex interactions between surface and subsurface flow (White, 2002). It is unclear, however, why this
interplay would lead to higher sensitivity to precipitation extremes and clustering.

Across Europe, differences in discharge sensitivity to precipitation extremes and to clustering, beyond the likely influence of snow discussed above, are less straightforward to interpret. The spatial coverage of the catchment ensemble is very uneven, which makes it difficult to identify robust spatial patterns. A more detailed analysis taking into account extreme precipitation





magnitude and seasonality, and clustering seasonality (Tuel and Martius, 2021a) is needed to better interpret our results.

We focused here on the link between precipitation clustering and high discharge. Still, whether high discharge translates into a flood, particularly a disastrous one, depends on many other factors, like antecedent catchment wetness, the presence of infrastructure and its management, or the timing of precipitation across the catchment (Merz et al., 2021). These factors can also interact across various spatial and temporal scales, depending on the flood event. The most disastrous floods tend to result from compounding effects among these various factors. Because disastrous floods remain quite rare, and may cover large areas, the

role played by TCEP in triggering such floods may be easier to quantify with cross-catchment analyses rather than by focusing on each catchment individually.

## 5  Conclusions

In this study, we quantified the effects of TCEP at sub-seasonal timescales on the occurrence and temporal persistence of high discharge in Switzerland and Europe. Our results across Europe confirm those of Tuel and Martius (2021b) for Switzerland:

clustering leads to a larger and more persistent discharge response, thus increasing the likelihood of high discharge compared to extreme precipitation events occurring in isolation. In addition, temporal clustering plays an important role in triggering periods of particularly persistent high discharge. These conclusions apply to the majority of analysed catchments, though catchment sensitivity to clustering varies with area, precipitation magnitude and discharge regimes. Despite its uneven spatial coverage across Europe, GRDC data agrees with the results obtained for Switzerland only. Compared to the Swiss data, it also allows to

assess how large catchments ($>10000$ km$^2$) respond to TCEP. A next step could be to analyse spatial patterns in GRDC results. By classifying precipitation extremes according to their timescale of clustering, we also find that clustering appears to be most relevant for high discharge at the 1-2 week timescale, beyond which its influence decreases. TCEP is therefore a critical driver of the occurrence and persistence in high discharge across all studied regions. Key for risk mitigation is thus to improve our understanding of where and why TCEP is likely to occur. We focused here on high and extreme discharge values. However,

very extreme discharge is by definition rare, and catchment-scale analyses may fail to select a sufficient number of events to obtain statistically significant links to TCEP. Starting instead from historical flood events may help to highlight how TCEP modulates extreme discharge and flood risk in a more robust way.

*Code availability.*  The R code to reproduce the results of this study is available at https://github.com/Quriosity129/Clustering_streamflow.

*Data availability.*  RhiresD is provided by MeteoSwiss, the Swiss Federal Office of Meteorology and Climatology. Switzerland discharge

data were obtained from Switzerland's Federal Office for the Environment. The EOBS dataset (Haylock et al., 2008) can be downloaded from https://cds.climate.copernicus.eu/cdsapp#!/dataset/10.24381/cds.151d3ec6?tab=overview (last accessed October 14, 2021). We acknowledge the Global Runoff Data Centre, 56068 Koblenz, Germany for making available GRDC data at https://portal.grdc.bafg.de/ (last accessed





March 9, 2021). GTOPO30 data (https://www.dx.doi.org/10.5066/F7DF6PQS) can be downloaded through the EarthExplorer at https://earthexplorer.usgs.gov/ (last accessed March 9,2021).

*Author contributions.* A.T. designed the research, implemented the code, analysed the data and provided the figures; O.M. designed and supervised the research; all authors discussed the results and contributed to the manuscript.

*Competing interests.* B.S. is a member of the editorial board of Hydrology and Earth System Sciences.

*Acknowledgements.* The authors gratefully acknowledge the Swiss Federal Office of the Environment (FOEN) and Regula Mülchi for the Swiss river discharge data. O.M. acknowledges support from the Swiss Science Foundation grant number 178751. J.Z. acknowledges the
Swiss National Science Foundation (grant no. 179876) and the Helmholtz Initiative and Networking Fund (Young Investigator Group COMPOUNDX; grant agreement no. VH-NG-1537).



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





**Table 1.** Distribution of extreme precipitation events across clustering categories in the Swiss (RhiresD/FOEN) and European (EOBS/GRDC) data: number of events averaged across all catchments and corresponding percentage (in brackets) relative to total number of events.

| Dataset | Non-clustered | 1 week | 2 weeks | 3 weeks | 4 weeks | 6 weeks | 8 weeks |
|---------|---------------|--------|---------|---------|---------|---------|---------|
| Switzerland | 93 (59) | 8 (5) | 11 (7) | 9 (6) | 8 (5) | 16 (10) | 13 (8) |
| Europe | 88 (58) | 9 (6) | 10 (7) | 9 (6) | 8 (5) | 15 (10) | 13 (8) |


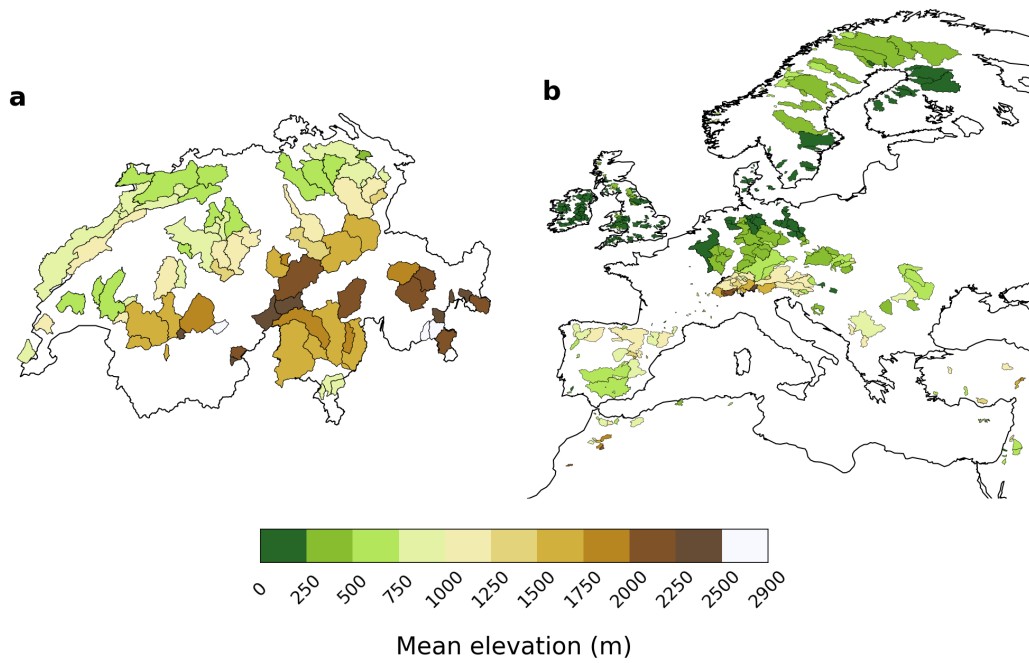

**Figure 1.** Map of (a) FOEN catchments across Switzerland and (b) GRDC catchments across Europe analysed in this study. Shading indicates mean catchment elevation.



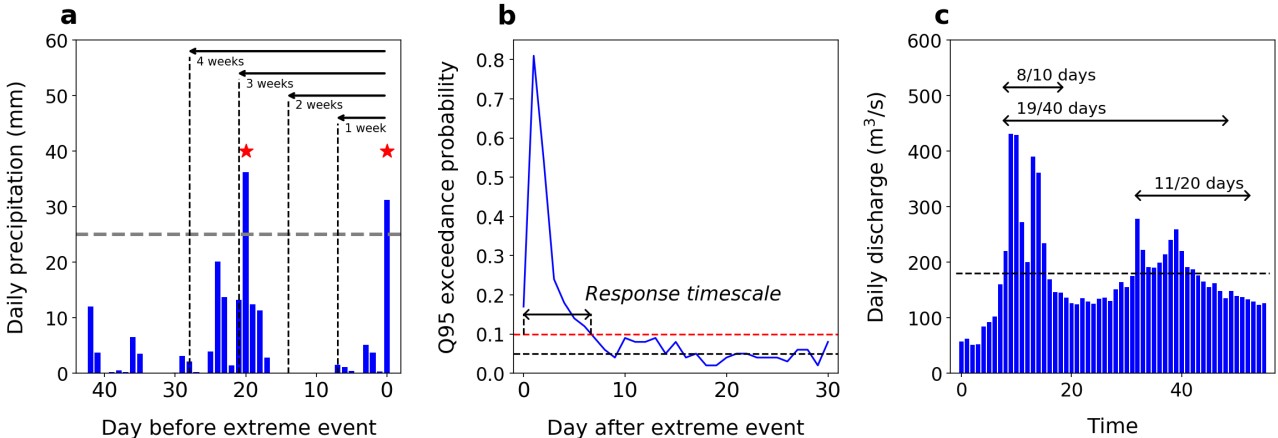

**Figure 2.** Metrics used to quantify the effects of TCEP on high discharge. (a) Clustering categories for precipitation extremes illustrated on an example daily precipitation time series (blue). Extreme events (above the $99^{th}$ daily percentile, horizontal dashed line) are indicated by red stars. To determine the clustering category for the event at t=0, we look for antecedent events in progressively longer time windows (1, 2, 3, 4, 6 and 8 weeks) (section 2.2.1) and choose the smallest window containing another extreme event. In this case, it would be the 3-week window. (b) Illustration of the definition of discharge response timescale (section 2.2.2). (c) Identification of persistent high discharge periods (section 2.2.3) illustrated on an example daily discharge time series (blue). Moving windows of various lengths $L$ are applied to select periods with a minimum number $N$ of high discharge days (larger than the $95^{th}$ daily percentile, horizontal dashed line): in order, $(L, N)$ equal to $(40, 20)$, $(20, 10)$, $(10, 5)$ and $(10, 1)$. Here, the two periods with 8/10 and 11/20 extreme days would be selected.

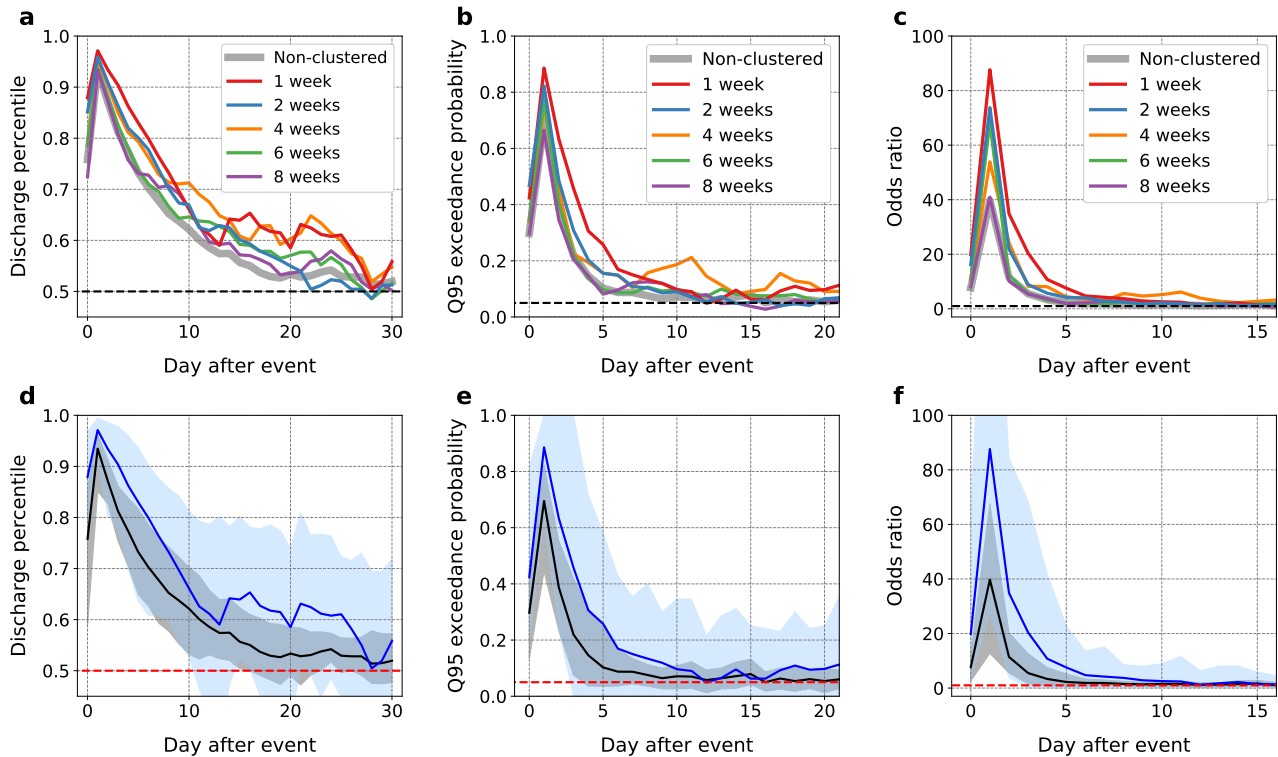

**Figure 3.** Daily average (a) discharge percentile, (b) probability of high discharge (defined as the exceedance of the respective $95^{th}$ daily discharge percentile) and (c) odds ratio of high discharge, averaged across FOEN catchments with a mean elevation of 1500m or less, for the different clustering categories of extreme precipitation. Black dashed lines indicate baseline values of 0.5 for discharge percentiles in (a), 0.05 for high discharge probability in (b) and 1 for odds ratios in (c). (d-f) Same as (a-c), but for the non-clustered (black) and 1-week clustered (blue) categories only, with the 95% range of values across catchments shown in light grey and blue shadings, respectively. Baseline values are shown by horizontal red dashed lines as in (a-c).





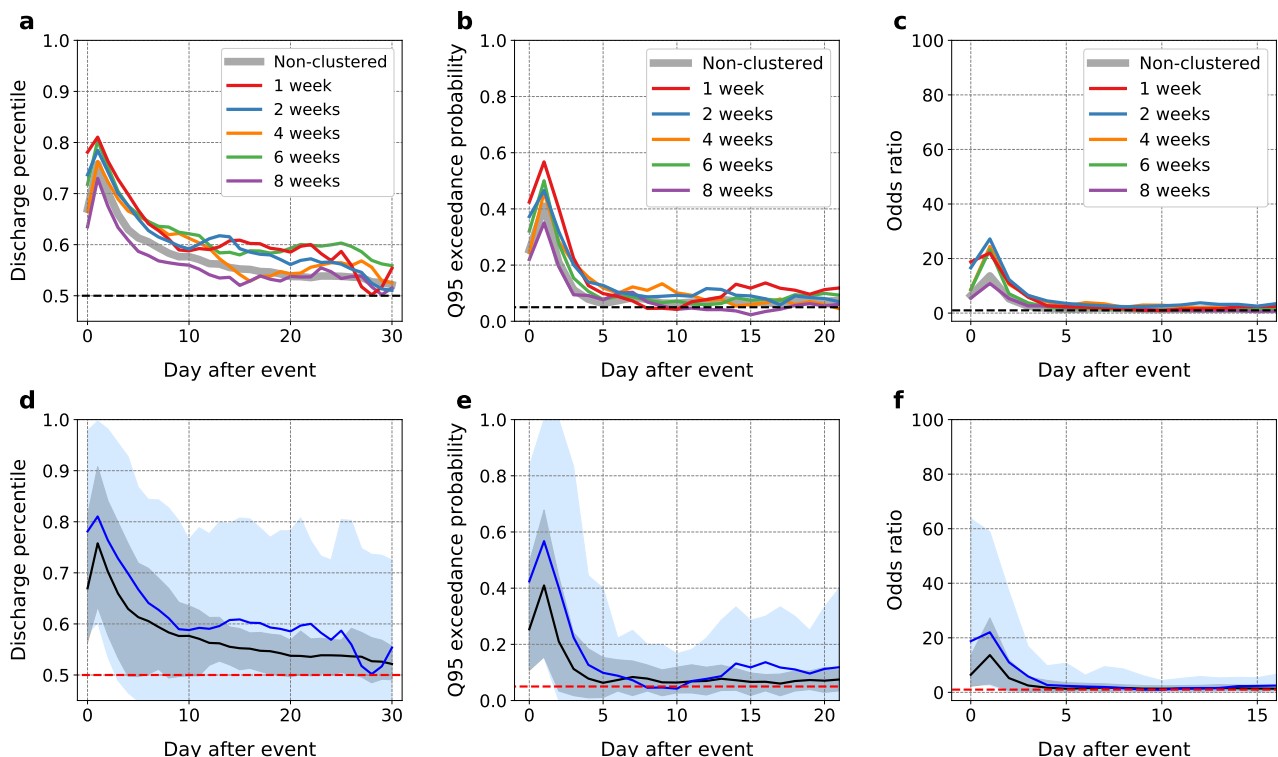

**Figure 4.** Same as Fig. 3, but for FOEN catchments with a mean elevation of more than 1500m.

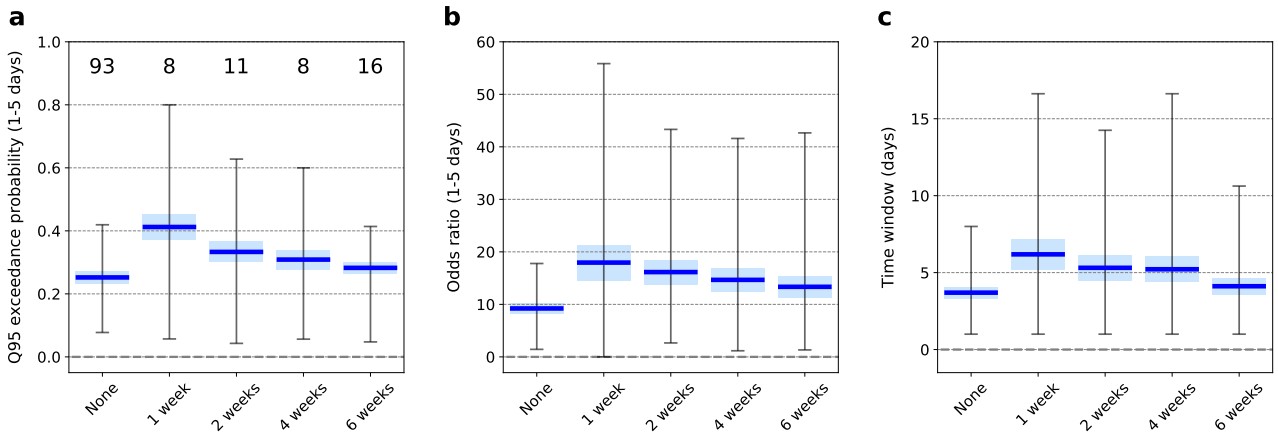

**Figure 5.** Boxplot of (a) high discharge probability and (b) high discharge odds ratio averaged over day 1-5 following the occurrence of an extreme precipitation event (day 0) for all FOEN catchments and various clustering categories. Numbers at the top in (a) indicate the average number of extreme events in the respective categories. (c) Boxplot of response timescale, defined in section 2.2.2, for all FOEN catchments and various clustering categories.

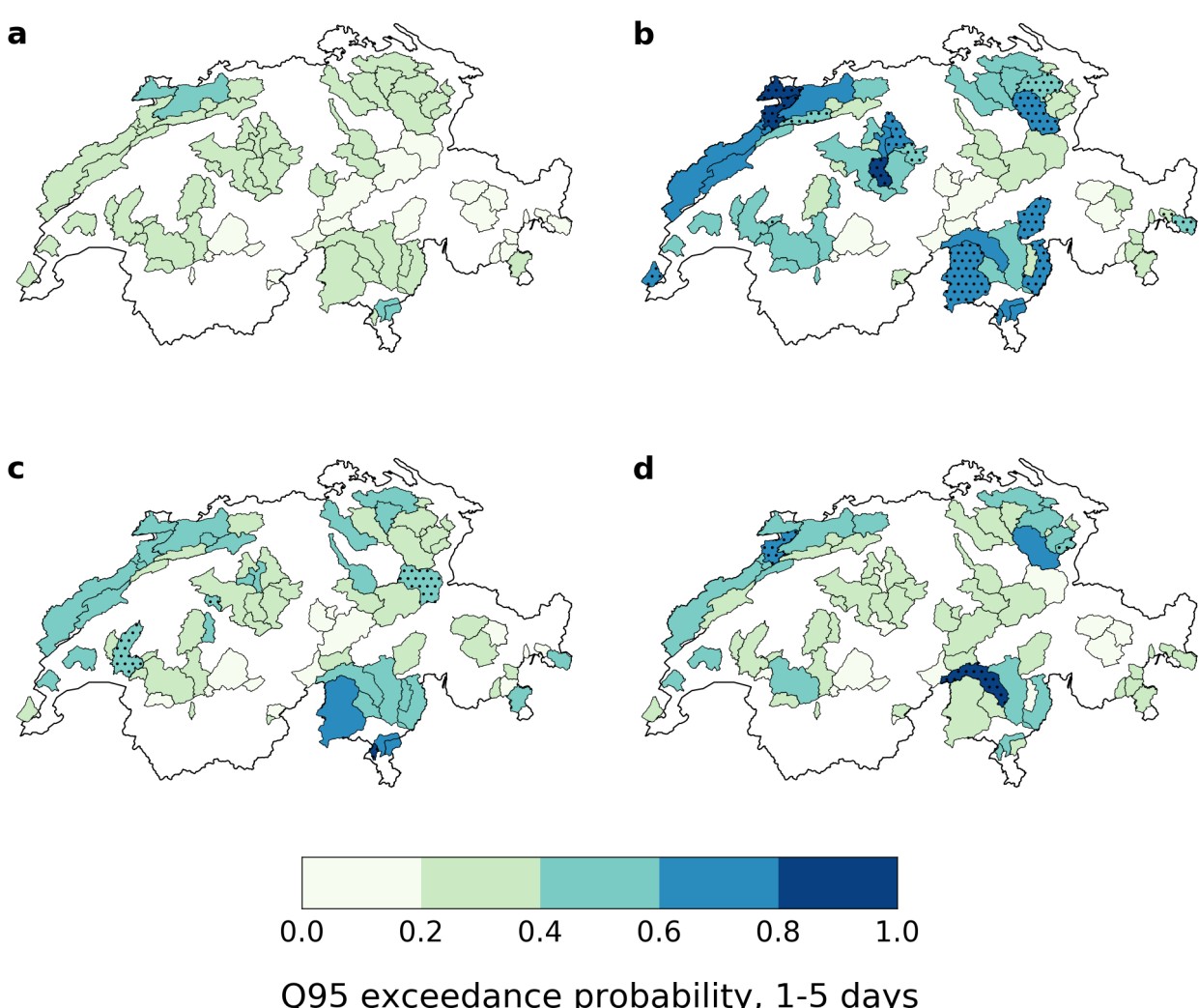

**Figure 6.** Average high discharge probability in day 1-5 following an extreme precipitation event, for (a) non-clustered, (b) 1-week clustered, (c) 2-week clustered and (d) 4-week clustered events, in the Swiss data. Hatching in (b-d) indicates catchments where values are significantly different from those in (a) at a 10% level.



**Figure 7.** Average high discharge odds ratio in day 1-5 following an extreme precipitation event, for (a) non-clustered, (b) 1-week clustered, (c) 2-week clustered and (d) 4-week clustered events, in the Swiss data.



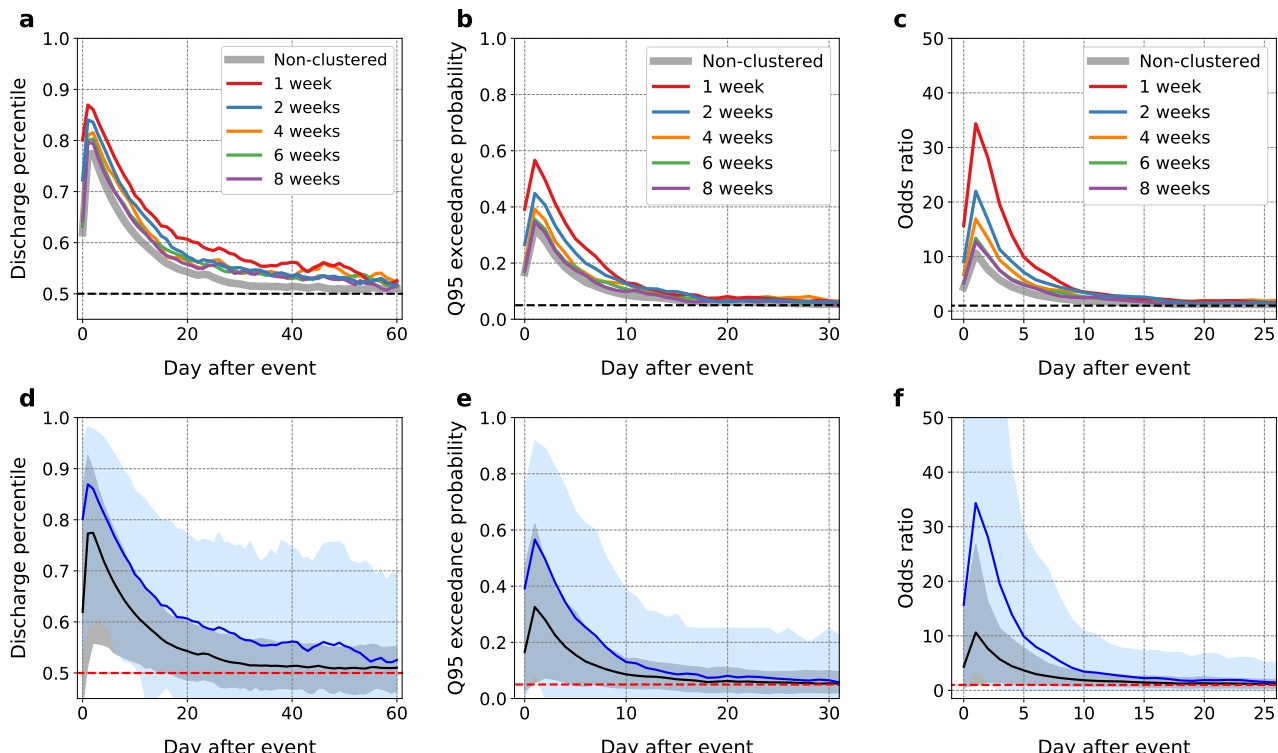

**Figure 8.** Same as Fig. 3, but for the European (EOBS/GRDC) data.





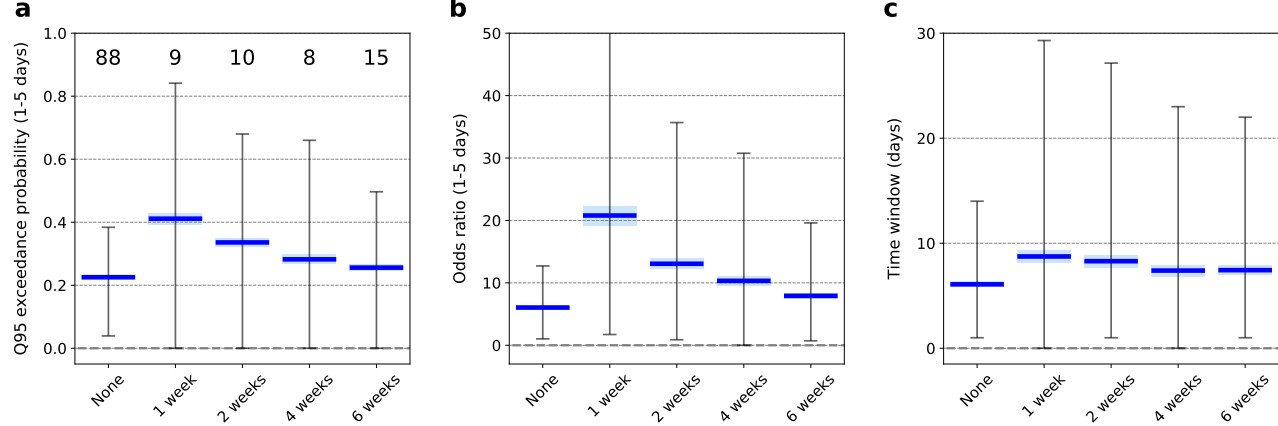

**Figure 9.** Same as Fig. 5, but for the European data.





**Figure 10.** Same as Fig. 6, but for the European data.



**Figure 11.** (a-f) Average percentile of cumulative precipitation (a-c) during day 0-2 before persistent high discharge periods and (d-f) during the persistent high discharge periods, and (g-i) average TCEP frequency from day 2 before to the end of persistent high discharge periods, for various values of $(L, N)$: (a,d,g) (10,1), (b,e,h) (20,10) and (c,f,i) (40,20) (see section 2.2.3 for details) across Switzerland. In (a-f) gray shading corresponds to non-significant values with a 90% level of significance.





**Figure 12.** Same as 11, but for the European data.



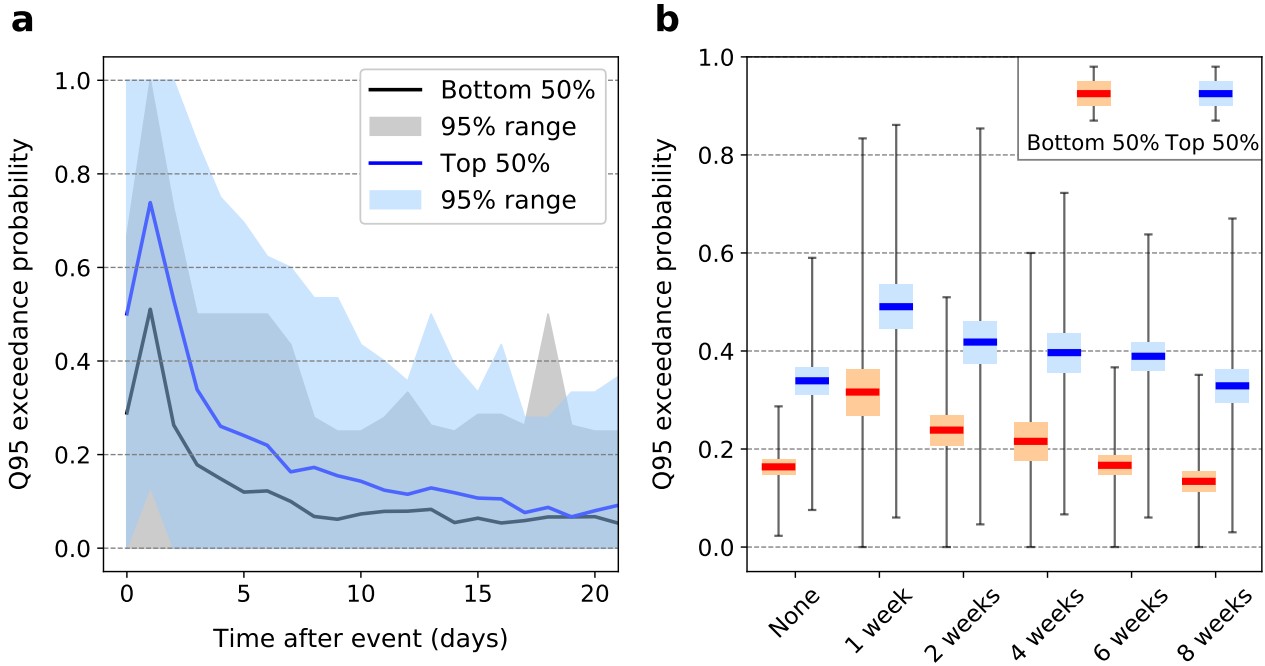

**Figure 13.** (a) Daily probability of high discharge following an extreme precipitation event (both clustered and non-clustered) averaged across Swiss catchments, separated between intense (top 50%, blue) and weak (bottom 50%, black) events based on their absolute magnitude. The 95% range of values across catchments is shown in light blue and black shadings, respectively. (b) Boxplot of high discharge probability averaged over day 1-5 following the occurrence of an extreme precipitation event (day 0), separated between intense and weak extremes, for various clustering categories, averaged over all Swiss catchments.



**Figure 14.** Daily average (a) discharge percentile, (b) probability of high discharge and (c) odds ratio of high discharge, averaged across "small" (area≤10,000 km$^2$, solid lines) and "large" (area≥10,000 km$^2$, dashed lines) catchments in the European data, for non-clustered (black), 1-week (blue) and 2-week (red) clustered events. (d) Average high discharge odds ratio from day 0-1 (blue) and day 4-5 (red) following an extreme precipitation event against catchment area.

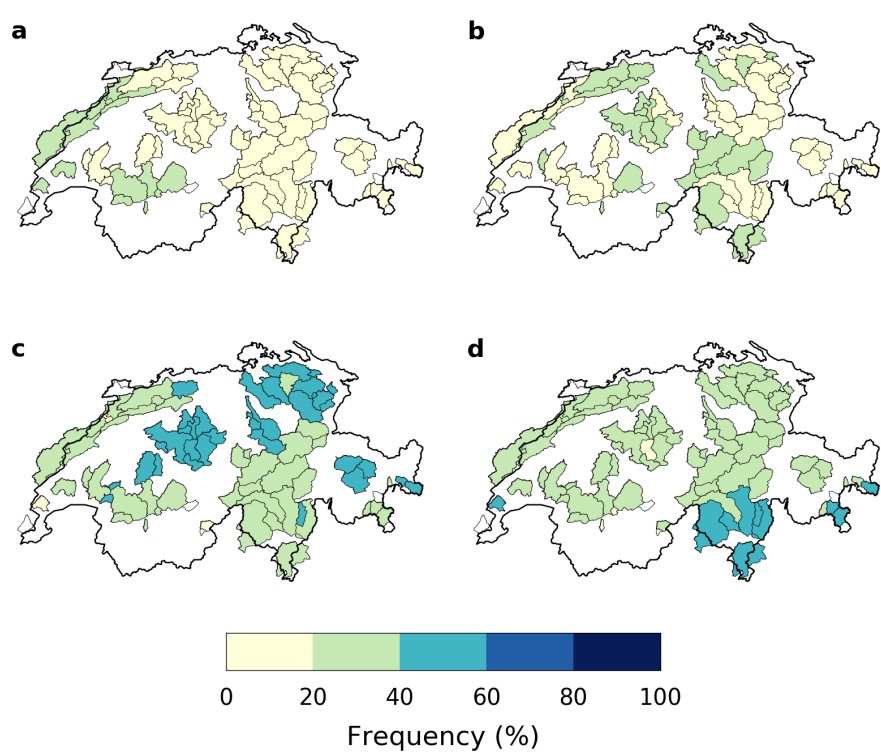

**Figure A1.** Seasonal frequency of exceedance of annual 99[th] daily precipitation percentile in Switzerland (RhiresD/FOEN data): (a) DJF, (b) MAM, (c) JJA and (d) SON.





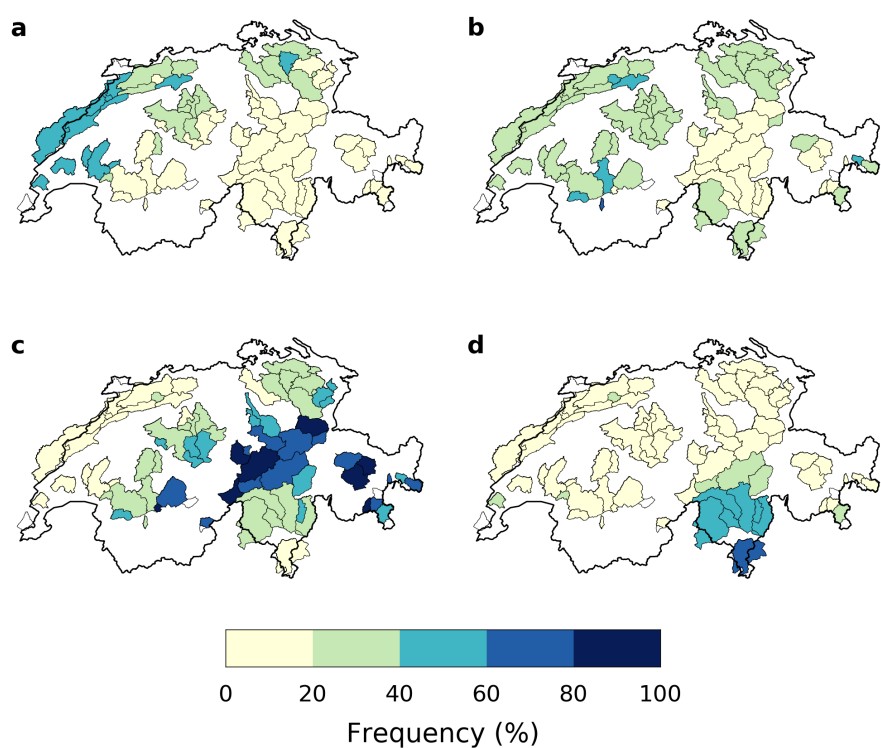

**Figure A2.** Seasonal frequency of exceedance of annual 95[th] daily discharge percentile in Swiss catchments: (a) DJF, (b) MAM, (c) JJA and (d) SON.

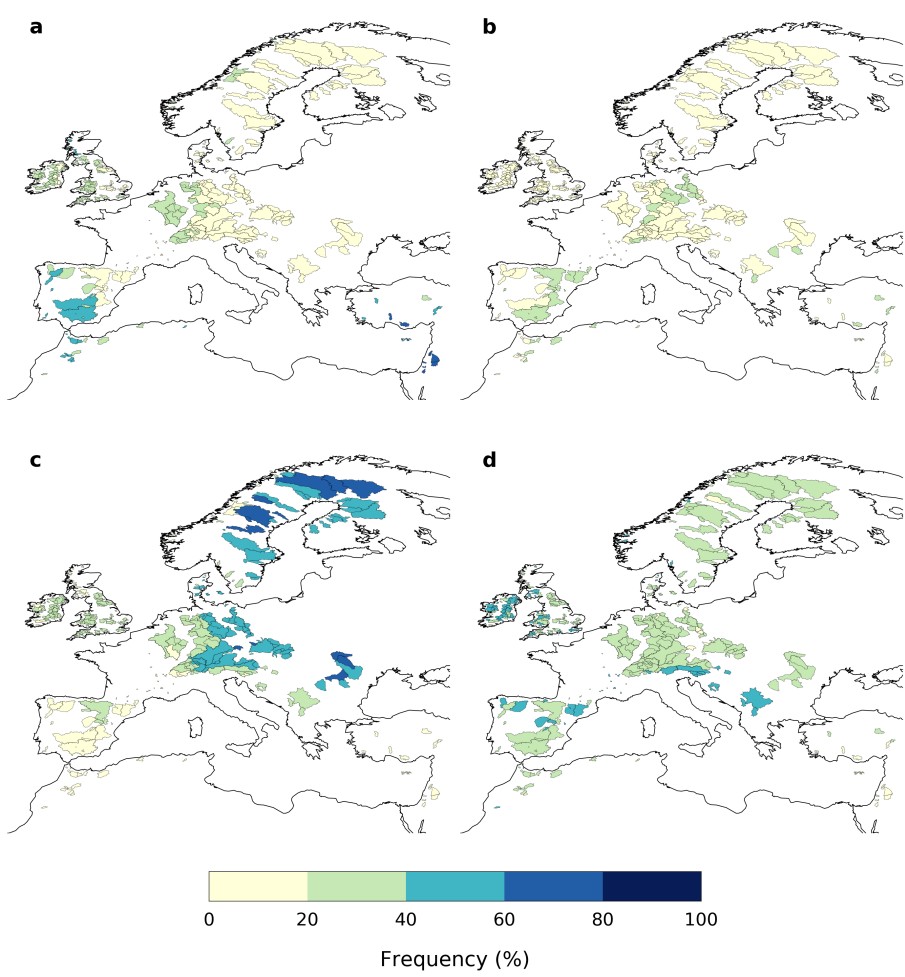

**Figure A3.** Seasonal frequency of exceedance of annual 99[th] daily precipitation percentile in Europe (EOBS/GRDC data): (a) DJF, (b) MAM, (c) JJA and (d) SON.



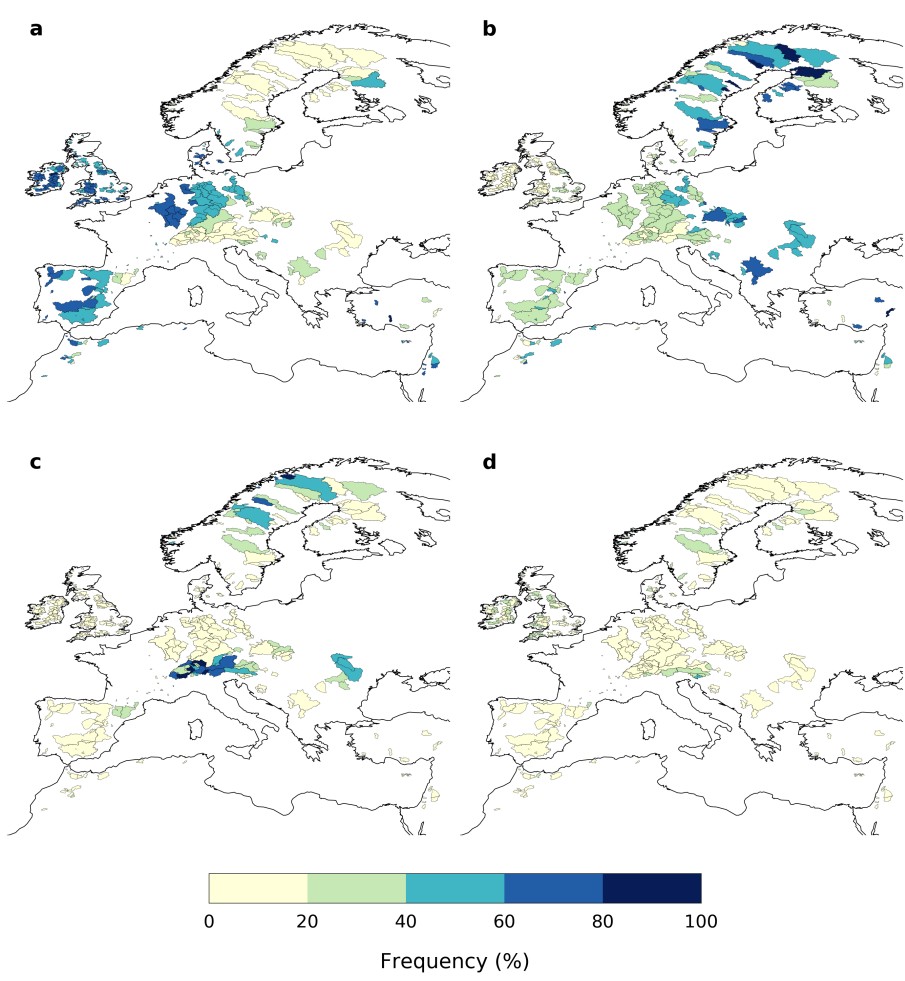

**Figure A4.** Seasonal frequency of exceedance of annual 95$^{th}$ daily discharge percentile in European catchments: (a) DJF, (b) MAM, (c) JJA and (d) SON.

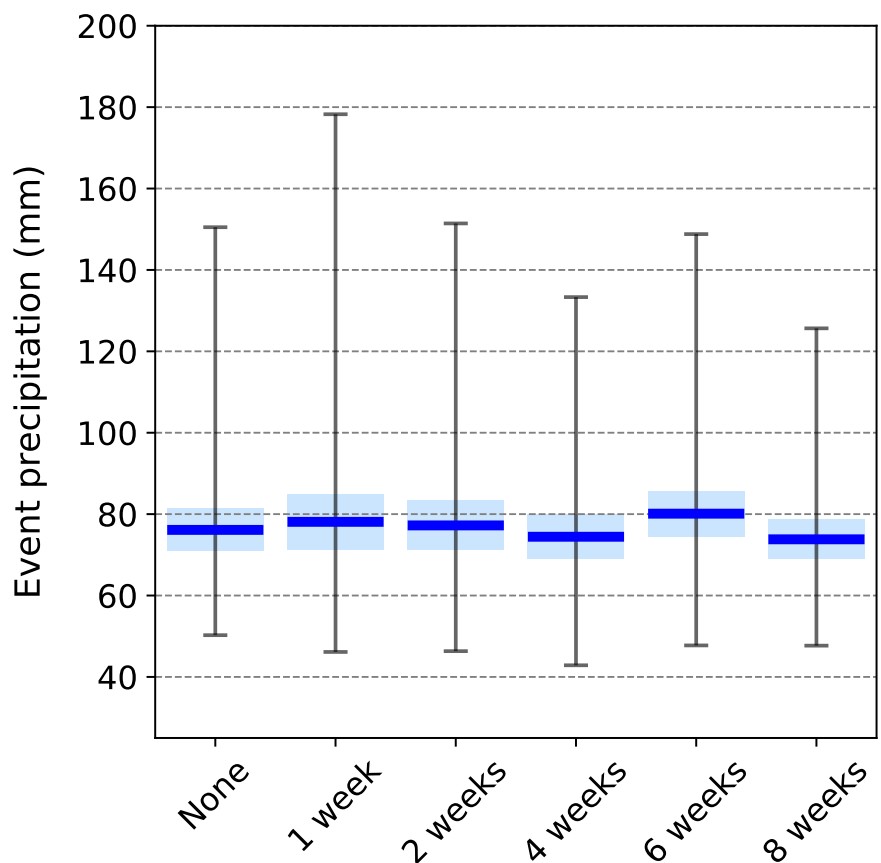

**Figure A5.** Boxplot of extreme precipitation event magnitude (total precipitation from one day before to one day after the event) as a function of clustering category in the Swiss data.

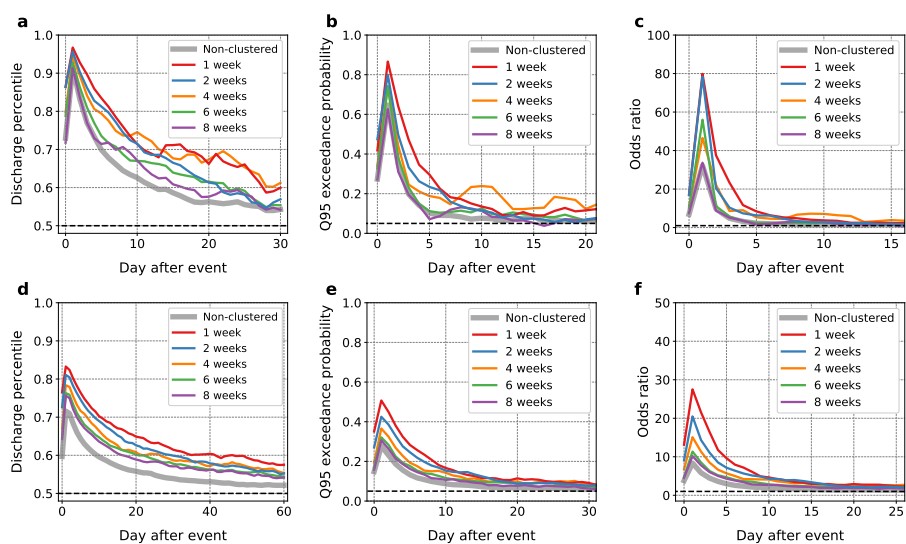

**Figure A6.** Daily average (a) discharge percentile, (b) probability of high discharge (defined as the exceedance of the respective 95[th] daily discharge percentile) and (c) odds ratio of high discharge, averaged across Swiss catchments with a mean elevation of 1500m or less, for the different clustering categories of extreme precipitation. Here the original discharge data (baseflow not removed) was used. Black dashed lines indicate baseline values of 0.5 for discharge percentiles in (a), 0.05 for high discharge probability in (b) and 1 for odds ratios in (c). (d-f) Same as (a-c), but for the European data.