# Peer review of "On the links between sub-seasonal clustering of extreme precipitation and high discharge in Switzerland and Europe"

_Hydrology and Earth System Sciences, 2021_

## Referee Comment (RC3)

**General comments**

The manuscript assesses the influence of temporal clustering of precipitation on discharge response. This is assessed with a forward and backward approach, for catchments in Switzerland and Europe. Temporal clustering of precipitation is an important factor in the generation of several natural hazards and a proper understanding of its influence on flood hazard, and natural hazards in general, is still lacking. In this regard, the manuscript provides novel and interesting results.

My main comment regards the identification of precipitation clusters:

- l.110 You identified clusters of precipitation events with a pairwise approach, categorizing each event depending on the distance in time from the previous closest event. The whole precipitation cluster is not identified, and consequently some events belonging to different categories may in reality be dependent, being part of the same precipitation cluster. I imagine that some events in a specific category may be preceded by other precipitation events and the whole story may influence the final discharge characteristics, rather than only the last two events of the cluster. Does this occur in your datasets? How much do you think this may affect the results?

- l.99 and l.114 Have you tried also lower precipitation quantiles or different pairs of precipitation and discharge quantiles? Are the results sensitive to this choice? Also (l.117) I do not think to be meaningful to chose discharge quantile starting from the chosen quantile of single precipitation events. Mainly considering that you show that extreme discharge is more often caused by precipitation events close in time than isolated.

Regarding the structure of the manuscript, I find it in general well-structured, with some exceptions reported in the technical comments below.

**Specific comments**

- l.77 Why did you not operate the same selection of catchments used for the Switzerland dataset (human influence, lakes, stationarity of the series . . . )?

- l.145-147 From this section it seems that all combinations of periods here reported are considered as persistent high discharge periods, included the pair (10, 1). However, also reading l.235, this is a non persistent period, right?

- Fig.11 I did not understand what is reported with cluster frequency.

- l.250 Why did you not use a measure, normalized for L, for example, that is comparable between the different pairs of (N, L)?

- l.293 Have you considered also the possibility of using rainfall rather than total precipitation?

- Fig. 13 Why did you choose an absolute precipitation magnitude rather than a magnitude relative to the quantile in each specific catchment (i.e. the excess over the threshold)?

**Technical corrections**

- l.69 *these catchment* → these catchments

- l.70 *The data is* → The data are

- l. 71 *We selected catchments among all available ones based on several criteria* → I would rewrite it like this: "Among all available catchments, we selected a subset of them based on several criteria:"

- l.76 *Daily discharge data for Europe comes from the Global Runoff Data Center dataset (GRDC).* → The second dataset consists of daily discharge data for Europe and it comes from the Global Runoff Data Center dataset (GRDC).

- l.92 *yielding two precipitation datasets* → I would remove this. You had two precipitation datasets also before.

- l.109 *events which occurred between n-1 and n weeks after another event are put into the "n-week" category, where n ∈ {1, 2, 3, 4, 5, 6, 7, 8}.* → I would substitute *another event* with *the previous extreme event*, this to make it clear that you are selecting the smallest window, or otherwise rewrite it more similarly to the explanation in the caption of Fig. 2.

- l.122 I think this part not to be reported clearly. In particular, I find misleading that you write that the probability is averaged across all extreme precipitation events belonging to a clustering category. Also, I would explicitly specify that the days (60 and 30 days) are the ones after each precipitation event. This is what I would probably write, but feel free to write it differently:

  *For each catchment, clustering category, and for each of the 30 (Switzerland data) or 60 (GRDC data) days following extreme precipitation events, we calculate*

  1. *daily discharge percentiles, averaged across all extreme precipitation events in each clustering category;*
  2. *daily high discharge probabilities;*
  3. *daily high discharge odds ratios.*

- Sec. 3.1.1 I believe that it is not always clear if the comments reported refer to all the catchments or only low/high elevation ones. In ll 172-187, for example, you are referring only to Fig. 3, that collects results of low elevation catchments but at the same time you are comparing it with Fig. 5, that, if I understood correctly, reports results for both subsets.

---

## Author Comment (AC1)

**Reviewer 1 comments**

**Comment 1.1** *The manuscript displays an interesting analysis of the impact of precipitation clustering on the duration and frequency of high discharge events in Switzerland and Europe. It uses appropriate data sets and makes a courageous attempt to stratify the analysis along the large number of degrees of freedom that govern the relationship between (clustered) extreme precipitation and high discharge. Overall this analysis and stratification are useful, but the authors seem to address the topic from a pretty methodological point of view, thereby presenting quite a large number of figures and results that are not always directly meaningful to the reader. In particular the large number of spatial maps don't convey a very clear message of spatial structure in the findings, and some reduction in figures and a more concise display of results would be appreciated. On the other hand, some potentially relevant physical processes are discussed somewhat in the discussion section and annex figures, and some of these would have been interesting to elaborate in somewhat more detail. Particularly the notion of (seasonally dependent) soil moisture memory and the impact of catchment size deserve a more explicit discussion and interpretation of the results.*

**Answer**: Thank you for your positive comments. We reduced the number of figures and simplified some of the spatial maps to narrow the message of the paper. Regarding the potential seasonal dependence of results, we slightly expanded the discussion in the manuscript and enclose a figure in this response. A more detailed seasonal analysis would however take us a bit far and make the paper even more complex.

**Comment 1.2** *L30-31: also Khanal et al (2019) address the role of atmospheric clustering and soil moisture memory on Rhine discharge explicitly*

**Answer**: Thank you for this reference. We suggest modifying the sentence as follows: "*To our knowledge, the impact of sub-seasonal TCEP on discharge has not been explicitly investigated, except briefly in the case of Switzerland by Tuel and Martius (2021b) and of the Rhine river basin by Khanal et al. (2019). Both argued that TCEP increased the likelihood and duration of high discharge events compared to precipitation extremes occurring in isolation.*"

**Comment 1.3** *L49-50: Kew et al (2013) looked at the effect of clustering on the probability to have compounding discharge and coastal surge peaks for the Rhine river. This is an interesting application domain of studying temporal clustering/compound events in this context*

**Answer**: This is an interesting example indeed involving coastal floods. Bevacqua et al. (2019) also analyse this question: https://dx.doi.org/10.1126/sciadv.aaw5531. We will add the reference in the introduction.

**Comment 1.4** *L81: given the relatively small effect of removing this baseflow component (L265-269), and the question of whether the resulting runoff data can be interpreted well I would suggest to leave out this baseflow correction and work with total discharge instead*

**Answer**: This is a good suggestion, and we propose removing the baseflow correction in the revised version altogether.

***Comment 1.5*** *L101-102: maybe add explicitly that these factors display a seasonal cycle*

**Answer**: Good point – here is what we suggest: "*This step is motivated by the fact that high discharge is shaped not only by precipitation, but also by seasonally-dependent surface conditions like snow and vegetation cover, soil saturation or evaporative demand.*"
* * *
***Comment 1.6*** *L116-117: the fact that discharge impact is characterized more by absolute than anomalous discharge also applies to the impact of clustered precipitation on discharge. So I find this methodological inconsistency of using absolute or anomalous values not convincing.*

**Answer**: We define extreme precipitation and discharge thresholds for each catchment independently, as it would be difficult to choose impact-relevant, absolute thresholds for so many different catchments. In addition, the discharge response to extreme precipitation is very dependent of surface conditions, which explains why the annual cycles in extreme precipitation and discharge are not necessarily in phase (Figures A1-A4). Consequently, it is not inconsistent to use monthly-varying precipitation percentiles to identify extreme precipitation events. We could also have used fixed percentiles, and the results averaged across catchments are in fact not very different (see below Figure R1). However, results can differ substantially for individual catchments, especially ones where the seasonality in extreme precipitation and extreme discharge is not in phase, like the Jura. Choosing fixed percentiles can also make us miss clustered events that nonetheless bring large precipitation accumulations and can have large discharge responses.

[Figure]

Figure R1: (Fixed (annual) percentiles: compare with Figure 5 of the manuscript) Boxplot of (a) high discharge probability and (b) high discharge odds ratio averaged over day 1-5 following the occurrence of an extreme precipitation event (day 0) for Swiss catchments and various clustering categories. Numbers at the top in (a) indicate the average number of extreme events in the respective categories. (c) Boxplot of response timescale for Swiss catchments and various clustering categories.
* * *
***Comment 1.7*** *L117: "smaller" à "lower"*

**Answer**: Noted, thanks.
* * *
[Figure]

Q95 exceedance probability, 1-5 days

Figure R2: (Fixed (annual) percentiles: compare with Figure 6 of the manuscript) Average high discharge probability in day 1-5 following an extreme precipitation event, for (a) non-clustered, (b) 1-week clustered, (c) 2-week clustered and (d) 4-week clustered events, in the Swiss data. Hatching in (b-d) indicates catchments where values are significantly different from those in (a) at a 10% level.

**Comment 1.8** *L126: the 30 and 60 days are not clear to me, I compared them to the max 8 weeks of clustering and couldn't find the logical match. Please explain somewhat better*

**Answer**: These two time windows correspond to the maximum time horizon we analyse after extreme precipitation events. The discharge response after that cannot be statistically separated from background noise. But having a different value for the Swiss and European data is unnecessarily confusing. In practice a 60-day window works for both. Thus we suggest reformulating this paragraph to make the point clearer:

"*We quantify the effect of temporal clustering of precipitation extremes on discharge by considering several simple metrics. For each catchment and each clustering category, we calculate for each day following extreme precipitation events:*

1. *daily discharge percentiles averaged across all events in the corresponding clustering category;*

2. *daily high discharge probabilities;*

3. *daily high discharge odds ratios*

*In practice, we limit the analysis to 60 days after extreme precipitation events, beyond which we do not find a significant discharge response.*"
* * *
**Comment 1.9** L147: what motivated this combination of N and L? It came somewhat unannounced.

**Answer**: The idea here is to look at sub-seasonal timescales (hence L between 10 and 40 days), and to set the threshold for "persistent" high discharge at 50% of high discharge days within a given period. To make this all clearer we suggest reformulating the beginning of the paragraph as follows: "*Following Tuel and Martius (2021b), we identify periods of persistent high discharge at sub-seasonal timescales as periods of 10 to 40 days when discharge exceeds its $95^{\text{th}}$ percentile at least half of the time. In practice, we look for L-day periods with at least N high discharge days, with $(L, N) \in \{(10, 5), (20, 10), (40, 20)\}$. We also consider an additional category, $(L, N) = (10, 1)$, to characterise non-persistent high discharge events.*"
* * *
**Comment 1.10** L166: "less variability": less than what?

**Answer**: We meant less uncertainty compared to the clustered categories, but the sentence can be removed altogether.
* * *
**Comment 1.11** L208: please refer forward to discussion section when introducing "karst effects" (and in the discussion section: please explain in some detail what this effect is about)

**Answer**: We suggest adding the following detail to the discussion: "*The Jura is a region that shows strong karst effects (where soluble limestone rocks dominate, leading to high permeability and complex subsurface flows) [...].*"
* * *
**Comment 1.12** L222-224: this should be discussed in the section on Swiss results, not here

**Answer**: Good point, we will move it there.
* * *
**Comment 1.13** L225-228: also this feels that it belongs to the methods section, not to the results section

**Answer**: You are right; we suggest moving several sentences that explain how we tested for the influence of elevation and of precipitation magnitude to the methods section:
"*Finally, because the phase of the precipitation and its magnitude impact the discharge response to precipitation extremes, we also analyse the Swiss results as function of catchment elevation – a rough proxy for the influence of snow – and of extreme precipitation magnitude. We separate Swiss catchments into two groups (one with mean elevation below 1500m, the other above 1500m) and average the results for each group. Discharge in high-elevation catchments is typically snow- or glacier-dominated, and we expect the discharge response to precipitation extremes to differ with elevation. We do not investigate the influence of elevation in the European data; first, because it covers a much narrower range of elevations (only 10 catchments have a mean elevation*

*higher than 1500m); second, because mean elevations are less representative of the elevation distribution in larger catchments; and third, because unlike in Switzerland, the presence of snow is dictated by other catchment characteristics (chiefly latitude). We also explore the sensitive to the event magnitudes, for this we separate extreme precipitation events in each catchment into two groups based on their absolute magnitude (bottom and top half), and average the results across catchments, for each group separately.*"
* * *
**Comment 1.14** *L236-237: "Cumulative precipitation ... periods.": I don't know what you want to convey with this statement*

**Answer**: We meant to say that even though both persistent and non-persistent high discharge periods were preceded by high cumulative precipitation, the precipitation percentiles were even larger before persistent periods than non-persistent ones. We suggest reformulating as follows:
"*Most high discharge periods, whether persistent or not, are preceded by intense precipitation ($90^{\text{th}}$ percentile or higher) in the three preceding days (Fig. 11-a,b,c). Still, accumulated precipitation tends to be even larger before persistent periods than before non-persistent ones, except at high elevations. The difference is largest for the most persistent periods (compare panels a and c), especially in the Jura and Southern Switzerland where values larger than the $98^{\text{th}}$ percentile are found.*"
* * *
**Comment 1.15** *Fig 11: the set of panels don't convey a very clear message of a spatially meaningful structure. Also it takes me a long time to make up my mind of what is actually shown in the different rows and columns. Also: what does "non-significant" (grey shading) mean in these panels? I don't really understand the grey shading in all basins of panel 11-d. I feel this display of information is too extensive.*

**Answer**: Figures 11 and 12 were indeed too complex, especially since we hardly discussed the spatial variability of the results. We propose to replace Figure 11 with more simple boxplots that still convey most of the information (see Figure R3), and to move Figure 12 to supplementary (also as a boxplot).
Additionally, your comment made us realise that we had forgotten to explain how we assessed the significance of the results shown on Figure 11. For each catchment and (L,N) value, we obtain a number (say, $m$) of persistent high-discharge periods. We calculate our various metrics (cumulative precipitation percentile in days 0-2 preceding those periods, etc.) and then assess their statistical significance by comparing them to metrics calculated from 1000 randomly generated samples of $m$ periods of same length $L$ at about the same time of year ($\pm 20$ calendar days) as the actual persistent high discharge periods. We will add the following paragraph to the methods section:
"*The cumulative precipitation percentiles are calculated with respect to all periods of the same length within $\pm 20$ calendar days of observed persistent high discharge periods. Their statistical significance is assessed with a Monte-Carlo approach. For each catchment and $(L, N)$ category, assuming we observe m periods of persistent high discharge, we generate 1000 random samples of m periods occurring within $\pm 20$ calendar days of observed high discharge periods, calculate cumulative precipitation percentiles for these random periods and obtain their $90^{\text{th}}$ percentile. Observed percentiles are then said to be significant if they exceed this value.*".

[Figure]

Figure R3: (a) Average percentile of cumulative precipitation during day 0-2 before and (b) during persistent high discharge periods, and (c) fraction of high discharge periods with two or more extreme precipitation events (TCEP) between day 2 before to the end of the period, averaged by catchment for the Switzerland data. Values are coloured according to catchment-mean elevation, to highlight the difference between high- and low-elevation catchments. In (a-b), triangles (resp. circles) indicate values that are (resp. are not) statistically significant at a 10% confidence level (see methods).
* * *
**Comment 1.16** *L237: "increase": of what relative to what?*

**Answer**: Please see our proposed reformulation of this sentence in your previous comment.
* * *
**Comment 1.17** *L245: events are classified by percentile, by persistence, by significance. It's not easy to disentangle all these attributes. What's the key point you want to convey here?*

**Answer**: The key point is that persistent high discharge periods are characterised by intense precipitation accumulations, unlike non-persistent periods. The statistical significance of the results is here secondary (more of a safety check, really). We can reformulate slightly to make it easier to read: "*Precipitation accumulations during high discharge periods are by contrast very different between persistent and non-persistent periods (Fig. 11-b). Small precipitation accumulations characterise non-persistent periods, whereas persistent periods are associated with high event precipitation totals except at high elevations.*"
* * *
**Comment 1.18** *L246: "often result from TCEP.": how can I deduce this statement from these panels? They look quite similar to panels 11d-f. And how does "cluster frequency" shown in Fig 11 translate to TCEP?*

**Answer**: The revised Figure 11 (Figure R3 here) shows the distribution of TCEP frequency during high discharge periods instead of spatial maps and should hopefully be easier to understand. Across catchments, we see for instance that 5-100% of persistent periods with (L,N)=(40,20) are associated with two or more precipitation extremes (i.e. TCEP), while these figures are only 1-7% for (L,N)=(10,1).

**Comment 1.19** *L249: "Overall, the connection to TCEP is weaker for less persistent high discharge periods.": how can I see this from the figure? What should I compare to what to understand this statement? Same for statement in L254*

**Answer**: Again, we hope that the revised Figure 11 (Figure R3) should be clearer.
* * *
**Comment 1.20** *L272-273: I didn't read the Bevacqua reference, but doesn't this combination of arguments imply that it is the change of the wet-day frequency and R95p that is responsible for this, without a significant change in clustering?*

**Answer**: Bevacqua et al. (2020) do find a significant change in cyclone clustering, namely that cyclone clusters will get shorter, which partly compensates for the increase in average precipitation associated with individual cyclones. Changes in clustering (of both signs) also matter for climate projections, even if in most regions the projected trends in extreme precipitation frequency dominate clustering trends (Tuel and Martius 2021a).
* * *
**Comment 1.21** *L274-275: this soil moisture memory plays a smaller role in winter/spring time, where the highest discharge occurs, I would reckon. A seasonal instead of a spatial analysis would have been more interesting (see also statement in L306)*

**Answer**: Soils indeed tend to be more saturated in winter so that soil moisture memory may be less important then. In summer, dry soils can nevertheless be less permeable and conducive to a large surface discharge response. The seasonality is worth exploring, which is challenging because it involves baseline discharge (Figures A1, A3), surface conditions and precipitation characteristics (Figure A2, A4). A detailed seasonal analysis by catchment is also difficult due to the potentially small number of clustered events (some clustering categories already have few events, so divided into four seasons they would be too few to say something remotely robust).

One way to tackle this problem is to simplify the clustering categories by grouping together all extreme events clustered at 1-4 weeks, and all others into a "non-clustered" category. This yields enough events for each catchment and category to get somewhat stable estimates of the various metrics. Results are shown on Figure R4 (on which we separated catchments based on elevation, since discharge in high-elevation catchments typically has a sharp peak in summer). As you suggested, the effect of clustering seems weaker in winter. Still, this remains a very rough analysis: as we said above, to reach robust conclusions, it would be necessary to simultaneously take into account spatial variability, seasonality in extreme precipitation and discharge, and variability in clustering timescales. But first, before even talking about clustering, one should detail the seasonality in the discharge response to extreme precipitation. This consequently goes beyond the scope of our study, which we see more as a first step to understanding the role of temporal clustering.

We suggest expanding the discussion section to bring up those points as follows: "*The role of the pre-conditioning through soil moisture is likely to vary across the year. In winter, soils are more likely to be saturated, so that the discharge response to small clustering windows may not be significantly higher. Yet, to explore the seasonality in TCEP impact on discharge, one would have to take into account seasonality in discharge and extreme precipitation magnitude (Figures S1-S4), in TCEP frequency (Tuel and Martius 2021a,b) and in surface conditions. All these factors make for a complex analysis which goes beyond the aim of the present study and would*

*likely require hydrological modeling, since at seasonal timescales clustered events might be too few to obtain robust statistical results.".*

[Figure]

Figure R4: (a-d) Daily average probability of high discharge (defined as the exceedance of the respective 95[th] daily discharge percentile) averaged across FOEN catchments with a mean elevation of 1500m or less, for the non-clustered and 1-4 week clustered categories of extreme precipitation, in (a) DJF, (b) MAM, (c) JJA and (d) SON. (e-h) Same as (a-d), but for FOEN catchments with mean elevation larger than 1500m.
* * *
**Comment 1.22** *L282: figures 11 and 12 don't show soil moisture impacts*

**Answer**: We suggest removing this sentence.
* * *
**Comment 1.23** *L307: "largest events": discharge or precipitation events?*

**Answer**: We meant extreme precipitation – we should reformulate by saying *"The heaviest extreme precipitation indeed generally occurs in summer and fall (Figure S1) [...]."*
* * *
**Comment 1.24** *L317: double use of "high"*

**Answer**: Thanks for noticing – we can replace by *"large high discharge probabilities persist for much longer."*
* * *
**Comment 1.25** *L324: 1000 km2 is pretty small. I would assume that clustered precipitation extremes can give high discharge in much larger basins*

**Answer**: There was a typo here, the figure should be 10000 km$^2$ (see Figure 14).
* * *
**Comment 1.26** *L330: "vary" à "varies"*

**Answer**: Noted, thanks.
* * *
**Comment 1.27** *L336: this seasonal signature is of great interest and could be promoted to the main text*

**Answer**: These maps are not a result of our study and would divert the reader away from the main point, since we do not explicitly analyse seasonal variations.
* * *
**Comment 1.28** *L351-352: both "antecedent soil moisture" and "timing of precipitation" refer to (temporal and spatial) clustering, so are attributes that are within scope of the current analysis*

**Answer**: You're correct and the wording was inadequate. We suggest "*Still, whether high discharge translates into a flood, particularly a disastrous one, depends on other factors related to the exposure and vulnerability of human systems, like the presence of infrastructure and its management, or the performance of early warning systems (Merz et al. 2021). The most disastrous floods tend to result from compounding effects between hazards, exposure and vulnerability.*"
* * *
**Comment 1.29** *L355: do "cross-catchment analyses" imply that you would show more results like fig 14? That would be great!*

**Answer**: The idea would indeed be to analyse several catchments with similar characteristics together. Currently, our cross-catchment analyses are rather coarse, since we average all available catchments together or, at best, divided into two groups based on elevation, which further research should refine.

---

## Author Comment (AC2)

**Reviewer 2 comments**

**Comment 2.1** *I think that this is a well-written and –structured study which highlights the importance of looking at precipitation events beyond the ones in immediate temporal proximity to high flow events. I have one methodological concern though regarding the event definition procedure and its potential impacts on the results/conclusions. In addition, I suggest to reconsider figure choice as the results of the Swiss and European analyses are very similar (one could maybe even say redundant). I think that narrowing down the selection might help to better focus the reader's attention on the main points.*

**Answer**: Thank you for your positive comments. We hope our responses will answer your concerns.

**Comment 2.2** *I suggest to clearly highlight the research gap after the paragraph ending on l.54. What remains to be investigated given the results presented in the authors' previous study?*

**Answer**: Thank you for this comment; we had not made it sufficiently clear in the manuscript what the novelty of the present study is. We suggest reformulating the last paragraph of the introduction as follows: *"How the impact of TCEP evolves with the timescale of clustering remains however unexplored. Each extreme precipitation event can in principle be associated with a clustering timescale, depending on the lapse of time since the previous extreme event. Tuel and Martius (2021b) only looked at 3-week clusters, analysing together extreme precipitation events at the beginning and end of the clusters. There is also interest in going beyond the borders of Switzerland, to consider a larger number of catchments with more diverse climates and less spatial dependence. Here, we quantify the effects of TCEP on discharge in Switzerland and Europe, specifically on the occurrence and temporal persistence of high discharge. We classify extreme precipitation events according to their clustering timescale, and analyse the sensitivity of results to that timescale, as well as to catchment area and to extreme precipitation magnitude."*

**Comment 2.3** *An overlap of only 10 years between precipitation and discharge events (l.72) seems very little given that the study focuses on extreme events.*

**Answer**: The thresholds we select are still low enough to capture a sufficient number of extreme precipitation (and high discharge) events (about 3-4 per year for precipitation, and 18 per year for discharge). Admittedly, some of the clustering categories may have very few events, but we take it into account to some extent by showing and discussing the uncertainty in and statistical significance of the results (Figures 6, 7 or 11 for instance).

**Comment 2.4** *Please specify how the baseflow filter parameter (l.82) was determined and why it does not vary in dependence of catchment properties.*

**Answer**: Following a comment by the previous reviewer, we suggest to show the results that include the baseflow since they do not differ much from results with baseflow removed (less so at high elevations).

**Comment 2.5** *Precipitation extremes are defined as anomalies (l.97-100) while discharge ex-*

*tremes are defined using a fixed threshold (l.115). While I understand the desire to consider precipitation events potentially co-occuring with wet antecedent conditions, I think this is not necessarily achieved by using a seasonally varying precipitation threshold (some events relevant in terms of discharge may still be missed). I think that choosing events based on the discharge rather than the precipitation events would be more consistent with the aim of the study. I think that the effect of choosing one over the other event identification procedure on the results of the analysis should be demonstrated in a small sensitivity analysis (e.g. on a small subset of catchments). Specifically, it would be important to know what effect (a) choosing events based on precipitation has compared to choosing events based on discharge and (b) what an effect choosing a seasonal rather than a fixed precipitation threshold has.*

**Answer**: Some precipitation events relevant for the discharge response are undoubtedly missed by our selection methods, but then this would also be the case with fixed (annual) thresholds, since in some catchments high discharge events do not always co-occur with the heaviest precipitation events. The point of the forward and backward approach is precisely to show that results are consistent whether we start from extreme precipitation events or from high discharge events. We can make this explicit in a revised version.

We chose monthly-varying percentiles to define extreme precipitation events because the discharge response to extreme precipitation is very dependent of surface conditions, which explains why the annual cycles in extreme precipitation and discharge are not necessarily in phase (Figures A1-A4). We could also have used fixed percentiles, and the results averaged across catchments are in fact not very different (see below Figure R1). However, results can differ substantially for individual catchments, especially ones where the seasonality in extreme precipitation and extreme discharge is not in phase, like the Jura (Figure R1). Choosing fixed percentiles can also make us miss clustered events that nonetheless bring large precipitation accumulations and can have large discharge responses.

[Figure]

Figure R1: (Fixed (annual) percentiles: compare with Figure 5 of the manuscript) Boxplot of (a) high discharge probability and (b) high discharge odds ratio averaged over day 1-5 following the occurrence of an extreme precipitation event (day 0) for Swiss catchments and various clustering categories. Numbers at the top in (a) indicate the average number of extreme events in the respective categories. (c) Boxplot of response timescale for Swiss catchments and various clustering categories.

[Figure]

[Figure]

Q95 exceedance probability, 1-5 days

Figure R2: (Fixed (annual) percentiles: compare with Figure 6 of the manuscript) Average high discharge probability in day 1-5 following an extreme precipitation event, for (a) non-clustered, (b) 1-week clustered, (c) 2-week clustered and (d) 4-week clustered events, in the Swiss data. Hatching in (b-d) indicates catchments where values are significantly different from those in (a) at a 10% level.
* * *
**Comment 2.6** *Were the percentiles computed empirically (l.155)?*

**Answer**: Yes, they were.
* * *
**Comment 2.7** *What about uncertainty in percentile computation because of small event sample sizes (l.159)? How does the analysis deal with cases where there are only very few events of the same length and what event sample sizes are we generally talking about?*

**Answer**: This comment made us realise that we had forgotten to explain how we assessed the significance of the results shown on Figure 11. For each catchment and (L,N) value, we obtain a number (say, $m$) of persistent high-discharge periods. We calculate our various metrics (cumulative precipitation percentile in days 0-2 preceding those periods, etc.) and then assess their

statistical significance by comparing them to metrics calculated from 1000 randomly generated samples of $m$ periods of same length $L$ at about the same time of year ($\pm 20$ calendar days) as the actual persistent high discharge periods. We will add the following paragraph to the methods section:

"*The cumulative precipitation percentiles are calculated with respect to all periods of the same length $\pm 20$ calendar days of observed persistent high discharge periods. Their statistical significance is assessed with a Monte-Carlo approach. For each catchment and $(L, N)$ category, assuming we observe $m$ periods of persistent high discharge, we generate 1000 random samples of $m$ periods occurring within $\pm 20$ calendar days of observed high discharge periods, calculate cumulative precipitation percentiles for these random periods and obtain their $90^{\text{th}}$ percentile. Observed percentiles are then said to be significant if they exceed this value.*".
* * *
**Comment 2.8** *I think that the European analysis is nice to put the Swiss analysis into broader perspective. However, I also think that it leads to the presentation of slightly too much material. I therefore suggest to just show a subset of the European results. I would try to focus the reader's attention to the most important information.*

**Answer**: We agree that the figures were too numerous and sometimes a bit too complex. We suggest replacing Figures 11 and 12 by more simple boxplots (since we do not discuss the spatial variability much in the end) and to move the revised Figure 12 to the supplementary material, since it is similar to the Swiss results.
* * *
**Comment 2.9** *l.3: 'this question' lacks a reference in the previous sentence.*

**Answer**: We can replace by "this topic", which refers to the "potential effects on discharge" (cf. "Its potential effects on discharge have received little attention. Here, we address this topic by analysing...").
* * *
**Comment 2.10** *l.10: the influence of temporal clustering on what? I understand that the statement refers to high flows and think that this could be made clearer.*

**Answer**: You are correct; we suggest reformulating as follows: "*The influence of temporal clustering on discharge decreases as the clustering window increases; beyond 6-8 weeks the difference in discharge response with non-clustered events is negligible.*"
* * *
**Comment 2.11** *l. 310: specify what 'this bias' refers to.*

**Answer**: We suggest "*Likewise, for some catchments, TCEP events occur in the season with the largest precipitation extremes, like in Southern Switzerland, which can bias the result since clustered events will also tend to be the heaviest.*"
* * *
**Comment 2.12** *Figure 1: suggest to reconsider color scale choice as a continuous variable is presented using a non-continuous color scheme.*

**Answer**: This is a standard color map for elevation, so we would prefer to keep it as it is.
* * *
**Comment 2.13** *Figure 2: I like figure 2. However, the horizontal lines in panel b should be*

**Answer**: You are right: "*The high discharge probability threshold of 0.1 is shown by the horizontal red line, and the baseline high discharge probability (0.05) by the horizontal black dashed line.*"
* * *
**Comment 2.14** *Figure 11: legend entry for grey color needed.*

**Answer**: The grey shaded catchments were the ones where the result was not statistically significant; in the revised Figure 11 this would be made explicit.

---

## Author Comment (AC3)

**Reviewer 3 comments**
* * *
**Comment 3.1** *l.110 You identified clusters of precipitation events with a pairwise approach, categorizing each event depending on the distance in time from the previous closest event. The whole precipitation cluster is not identified, and consequently some events belonging to different categories may in reality be dependent, being part of the same precipitation cluster. I imagine that some events in a specific category may be preceded by other precipitation events and the whole story may influence the final discharge characteristics, rather than only the last two events of the cluster. Does this occur in your datasets? How much do you think this may affect the results?*

**Answer**: You are correct that extreme precipitation events are categorised based on their distance in time to previous events only. For instance, an event occurring on day 100, 10 days after another event on day 90, would be put in the "2-week" category, but the event on day 90 may belong to a different category, depending on the time of the previous event. This choice is consistent with the physical perspective: the first event in a cluster conditions the discharge response to later events, but not the other way around. Thus, it doesn't make sense to automatically put the first event of a cluster in the same category as the later events.

If we did that, however, notwithstanding the physical inconsistency, we expect the results to be generally less significant and the discharge response after clustered events to be closer to the one after non-clustered events (since many events previously categorised as non-clustered would then be classified as clustered).
* * *
**Comment 3.2** *l.99 and l.114 Have you tried also lower precipitation quantiles or different pairs of precipitation and discharge quantiles? Are the results sensitive to this choice? Also (l.117) I do not think to be meaningful to chose discharge quantile starting from the chosen quantile of single precipitation events. Mainly considering that you show that extreme discharge is more often caused by precipitation events close in time than isolated.*

**Answer**: The results are qualitatively unchanged for lower precipitation percentiles (e.g. 95th percentile, compare Figures R1 and R2 with Figures 3 and 5 in the manuscript). The choice of a discharge percentile lower than that for precipitation is common in hydrology. It is meant to capture as many of the discharge extremes following extreme precipitation events as possible. Extreme discharge events are indeed not exclusively triggered by extreme precipitation events, and surface conditions in particular strongly modulate the discharge response to extreme precipitation. An extreme (>99th percentile) precipitation event therefore does not necessarily lead to a discharge response above the 99th percentile.
* * *
**Comment 3.3** *l.77 Why did you not operate the same selection of catchments used for the Switzerland dataset (human influence, lakes, stationarity of the series...)?*

**Answer**: We forgot to mention that we did test for the stationarity of the series in the GRDC dataset as well (by testing for significant trends in annual discharge maxima with a Mann-Kendall test; for the Swiss data, the stationarity analysis is performed by the FOEN). We would specify it in the revised version. As to the other criteria, however, the metadata of the GRDC

[Figure]

Figure R1: Daily average (a) discharge percentile, (b) probability of high discharge (defined as the exceedance of the respective 95th daily discharge percentile) and (c) odds ratio of high discharge, averaged across FOEN catchments with a mean elevation of 1500m or less, for the different clustering categories of extreme precipitation (exceeding its 95th percentile). Black dashed lines indicate baseline values of 0.5 for discharge percentiles in (a), 0.05 for high discharge probability in (b) and 1 for odds ratios in (c). (d-f) Same as (a-c), but for the non-clustered (black) and 1-week clustered (blue) categories only, with the 95% range of values across catchments shown in light grey and blue shadings, respectively. Baseline values are shown by horizontal red dashed lines as in (a-c).
* * *
dataset is unfortunately insufficient to determine which catchments strongly anthropogenically influenced or contain major lakes.
* * *
**Comment 3.4** *l.145-147 From this section it seems that all combinations of periods here reported are considered as persistent high discharge periods, included the pair (10, 1). However, also reading l.235, this is a non persistent period, right?*

**Answer**: You are correct and the initial formulation was misleading. We suggest reformulating as follows: "*Following Tuel and Martius (2021b), we identify periods of persistent high discharge at sub-seasonal timescales as periods of 10 to 40 days when discharge exceeds its 95*th *percentile at least half of the time. In practice, we look for L-day periods with at least N high discharge days, with $(L, N) \in \{(10, 5), (20, 10), (40, 20)\}$. We also consider an additional category, $(L, N) = (10, 1)$, to characterise non-persistent high discharge events.*"
* * *
**Comment 3.5** *Fig.11 I did not understand what is reported with cluster frequency.*

**Answer**: We should have said "TCEP frequency", meaning the fraction of high discharge periods with two or more extreme precipitation events.

[Figure]

Figure R2: Boxplot of (a) high discharge probability and (b) high discharge odds ratio averaged over day 1-5 following the occurrence of an extreme precipitation event (day 0, 95th percentile) for all FOEN catchments and various clustering categories. (c) Boxplot of response timescale for all FOEN catchments and various clustering categories.
* * *
**Comment 3.6** *l.250 Why did you not use a measure, normalized for L, for example, that is comparable between the different pairs of (N, L)?*

**Answer**: For persistent high-discharge periods, we already choose N=L/2 for all three categories (L=10, 20 or 40 days). We suggest making it explicit in the methods by adding "*Following Tuel and Martius (2021b), we identify periods of persistent high discharge at sub-seasonal timescales as periods of 10 to 40 days when discharge exceeds its 95$^\text{th}$ percentile at least half of the time.*
* * *
**Comment 3.7** *l.293 Have you considered also the possibility of using rainfall rather than total precipitation?*

**Answer**: The available gridded precipitation data does not discriminate between solid and liquid precipitation. Admittedly, we could use air temperature as a proxy, but only one value per day is available (and values in mountain areas where snow is most frequent are less reliable since stations are less numerous there). We prefer here to keep the analysis simple, even if it adds limitations, and explore the role of snow at first order only (Figures 3-4).
* * *
**Comment 3.8** *Fig. 13 Why did you choose an absolute precipitation magnitude rather than a magnitude relative to the quantile in each specific catchment (i.e. the excess over the threshold)?*

**Answer**: The selection of the two classes is done on each catchment separately, so we do use a relative precipitation threshold. To make it clear we suggest reformulating the corresponding sentence (l.301) as follows: "*A simple way to tackle this is to separate, for each catchment separately, extreme precipitation events into two groups based on their absolute magnitude.*"
* * *
**Comment 3.9** *l.69 these catchment → these catchments*

**Answer**: Corrected, thanks.
* * *
**Comment 3.10** *l.70 The data is → The data are*

**Answer**: Corrected, thanks.
* * *
**Comment 3.11** *l. 71 We selected catchments among all available ones based on several criteria → I would rewrite it like this: "Among all available catchments, we selected a subset of them based on several criteria:"*

**Answer**: Good suggestion, thanks.
* * *
**Comment 3.12** *l.76 Daily discharge data for Europe comes from the Global Runoff Data Center dataset (GRDC). → The second dataset consists of daily discharge data for Europe and it comes from the Global Runoff Data Center dataset (GRDC).*

**Answer**: Good suggestion, thanks.
* * *
**Comment 3.13** *l.92 yielding two precipitation datasets → I would remove this. You had two precipitation datasets also before.*

**Answer**: Good suggestion, thanks.
* * *
**Comment 3.14** *l.109 events which occurred between n-1 and n weeks after another event are put into the "n-week" category, where $n \in \{1, 2, 3, 4, 5, 6, 7, 8\}$. → I would substitute another event with the previous extreme event, this to make it clear that you are selecting the smallest window, or otherwise rewrite it more similarly to the explanation in the caption of Fig. 2.*

**Answer**: Good point; we suggest to reformulate as follows: *"For each catchment, we then classify precipitation extremes into different categories based on their degree of sub-seasonal temporal clustering (Figure 2-a). For each extreme event, we look for the previous event closest in time, by exploring progressively longer time windows of n weeks ($n \in \{1, 2, 3, 4, 5, 6, 7, 8\}$). We choose the first (i.e., shortest) window that contains the closest previous event."*
* * *
**Comment 3.15** *l.122 I think this part not to be reported clearly. In particular, I find misleading that you write that the probability is averaged across all extreme precipitation events belonging to a clustering category. Also, I would explicitly specify that the days (60 and 30 days) are the ones after each precipitation event. This is what I would probably write, but feel free to write it differently:*
*For each catchment, clustering category, and for each of the 30 (Switzerland data) or 60 (GRDC data) days following extreme precipitation events, we calculate*
*1. daily discharge percentiles, averaged across all extreme precipitation events in each clustering category;*
*2. daily high discharge probabilities;*
*3. daily high discharge odds ratios.*

**Answer**: Thank you for the comment. We suggest reformulating this paragraph as follows:
*"We quantify the effect of temporal clustering of precipitation extremes on discharge by considering several simple metrics. For each catchment and each clustering category, we calculate for each day following extreme precipitation events:*

1. *daily discharge percentiles averaged across all events in the corresponding clustering category;*

2. *daily high discharge probabilities;*

3. *daily high discharge odds ratios*

*In practice, we limit the analysis to 60 days after extreme precipitation events, beyond which we do not find a significant discharge response.*"
* * *
**Comment 3.16** *Sec. 3.1.1 I believe that it is not always clear if the comments reported refer to all the catchments or only low/high elevation ones. In ll 172-187, for example, you are referring only to Fig. 3, that collects results of low elevation catchments but at the same time you are comparing it with Fig. 5, that, if I understood correctly, reports results for both subsets.*

**Answer**: The original Figure 5 indeed grouped together both high- and low-elevation catchments. We suggest to replace it with one that separates the two groups of catchments:

[Figure]

Figure R3: Boxplot of (a) high discharge probability and (b) high discharge odds ratio averaged over day 1-5 following the occurrence of an extreme precipitation event (day 0) for FOEN catchments with mean elevation lower (blue) and higher (orange) than 1500 m, and various clustering categories. Numbers at the top in (a) indicate the average number of extreme events in the respective categories. (c) Boxplot of response timescale for FOEN catchments and various clustering categories.

---

## Author Response (AR2)

Dear Editor,

Thank you for your message. We are happy that the reviewers were satisfied with our revisions. We made the few corrections required to answer the reviewers' last remaining points, and are happy to send you our final manuscript.

Thanks again to you and to the two reviewers for your time and efforts.

Best wishes,

Alexandre Tuel (on behalf of all authors)

**Reviewer #1 comments**

Fig. 11 and Fig. A5: see methods → report the reference to the specific section of Methods.
Line 308: see methods → same as before
Fig. A5: the limits of panel (c) are not set correctly
Sec. 4.3: check figures' references
All points checked and corrected.

**Reviewer #2 comments**

l.21-22: The newly added sentence seems out of place/unrelated to topic and I would again remove it because it disturbs the reading flow.
Sentence removed.

l. 37: P is not usually used in flood frequency analysis and I think that this sentence needs rephrasing.
We rephrased as follows: "*The study of the influence of the temporal structure of precipitation on the catchment-scale hydrologic response is one of the foundations of runoff response analysis and flood frequency estimation.*"

L. 64: what are you referring to by spatial dependence?
We meant to say that Swiss catchments, being nearer in space, exhibited precipitation and discharge series that were more correlated in time. We rephrased the sentence as follows: "*There is also interest in going beyond the borders of Switzerland, to consider a larger number of catchments that cover more diverse climates and whose precipitation and discharge series are less correlated*".

L. 157: sensitivity to…
Corrected, thanks.

Section 4.3: quite a few undefined figure references.
Bug fixed.

L. 375: the role of discharge regime on clustering behavior has not been investigated in this catchment and I would rephrase this sentence.
We rephrased the sentence as follows: "*Despite its uneven spatial coverage across Europe, GRDC could be used to further analyse the sensitivity of the TCEP response to discharge regimes, and to detect potential spatial patterns.*"

L. 390: contributions of JZ and BS missing. Specify.
Their contributions are specified in "all authors discussed the results and contributed to the manuscript".

Number of figures still at the upper limit. Are all these figures really needed?
We moved figure 7 to the supplementary material.